# The impact of site-specific digital histology signatures on deep learning model accuracy and bias

Frederick M. Howard [1], James Dolezal[1], Sara Kochanny[1], Jefree Schulte [2], Heather Chen[2], Lara Heij [3,4], Dezheng Huo [5,6], Rita Nanda [1,6], Olufunmilayo I. Olopade [1,6], Jakob N. Kather[7,8,9], Nicole Cipriani [2,6], Robert L. Grossman[1,6 ✉] & Alexander T. Pearson [1,6 ✉]

The Cancer Genome Atlas (TCGA) is one of the largest biorepositories of digital histology. Deep learning (DL) models have been trained on TCGA to predict numerous features directly from histology, including survival, gene expression patterns, and driver mutations. However, we demonstrate that these features vary substantially across tissue submitting sites in TCGA for over 3,000 patients with six cancer subtypes. Additionally, we show that histologic image differences between submitting sites can easily be identified with DL. Site detection remains possible despite commonly used color normalization and augmentation methods, and we quantify the image characteristics constituting this site-specific digital histology signature. We demonstrate that these site-specific signatures lead to biased accuracy for prediction of features including survival, genomic mutations, and tumor stage. Furthermore, ethnicity can also be inferred from site-specific signatures, which must be accounted for to ensure equitable application of DL. These site-specific signatures can lead to overoptimistic estimates of model performance, and we propose a quadratic programming method that abrogates this bias by ensuring models are not trained and validated on samples from the same site.

---

[1] Section of Hematology/Oncology, Department of Medicine, University of Chicago, Chicago, IL, USA. [2] Department of Pathology, University of Chicago, Chicago, IL, USA. [3] Department of Surgery and Transplantation, University Hospital RWTH Aachen, Aachen, Germany. [4] Institute of Pathology, University Hospital RWTH Aachen, Aachen, Germany. [5] Department of Public Health Sciences, University of Chicago, Chicago, IL, USA. [6] University of Chicago Comprehensive Cancer Center, Chicago, IL, USA. [7] Department of Medicine III, University Hospital RWTH Aachen, Aachen, Germany. [8] Pathology and Data Analytics, Leeds Institute of Medical Research at St James's, University of Leeds, Leeds, UK. [9] Medical Oncology, National Center for Tumor Diseases, University Hospital Heidelberg, Heidelberg, Germany. ✉email: rgrossman1@uchicago.edu; apearson5@medicine.bsd.uchicago.edu

A standard component of the diagnosis of nearly all human cancers is the histologic examination of hematoxylin and eosin-stained tumor biopsy sections. Histologic characteristics identified by pathologists help characterize tumor subtypes, prognosis, and at times can predict response to treatment[1]. Quantification of more subtle pathologic features can further discriminate between good and poor prognosis tumors, such as the quantification of tumor-infiltrating lymphocytes in breast cancer, but such detailed analysis can be time-consuming and variable between pathologists[2]. The increasing availability of digital histology coupled with advances in artificial intelligence and image recognition has led to computational approaches to rigorously assess pathologic correlates associated with a variety of tumor-specific features. Deep learning is a subdomain of artificial intelligence, referring to the use of multilayer neural networks to identify increasingly higher-order image characteristics to allow for the accurate identification of features of interest. Deep learning on digital histology has exploded as a potential tool to identify standard histologic features such as grade[3,4], mitosis[5,6], and invasion[7,8]. Recently, deep-learning approaches have been applied to identify less apparent features of interest, including clinical biomarkers such as breast cancer receptor status[4,9], microsatellite instability[10,11], or the presence of pathogenic virus in cancer[12]. These approaches have been further extended to infer more complex features of tumor biology directly from histology, including gene expression[13–15] and pathogenic mutations[16,17]. The predictive accuracy of many of these models has been validated in external datasets, but studies often rely on single-data sources for both training and validation.

The Cancer Genome Atlas (TCGA) has been critical for the development of deep-learning histology models, containing over 20,000 digital slide images from 24 tumor types, along with associated clinical, genomic, and radiomic data[18]. Due to the propensity of machine learning algorithms to overfit, performance is typically reported in a reserved testing set or evaluated with cross-validation, to avoid biased estimates of accuracy[19]. However, the overfitting of digital histology models to site-level characteristics has been incompletely characterized and is infrequently accounted for in the internal validation of deep learning models. The genomic batch effects in TCGA and other high-throughput sequencing endeavors have been well -characterized, and are the product of the hundreds of tissue source sites contributing samples and the multiple sites for genome sequencing and characterization[20–22]. Histologic imaging data similarly contain characteristic signatures unique to each tissue submitting site (Fig. 1). Prior to sectioning, tissue is first fresh-frozen or fixed in formalin and embedded in paraffin, and each fixation method generates unique artifacts[23]. Slides are then stained with the eponymous hematoxylin and eosin stains, the color and intensity of which can vary based on the specific stain formulation and the amount of time each stain is applied. The digitization of slides may then vary due to scanner calibration and choice of resolution and magnification[24,25]. Finally, histologic characteristics of tumors can differ between institutions, due to biological differences between the patients treated at different centers. Thus, differences in specimen acquisition, staining, digitization, and patient demographics all contribute to a unique site-specific digital histology signature, which could in turn lead to a lack of generalizability of digital imaging models.

Several methods have been proposed to eliminate these site-specific signatures to improve the validity of histologic image analysis, primarily through correcting for differences in slide staining between institutions[26]. This includes methods designed to reduce color variation across images proposed by Reinhard et al.[27], and methods designed specifically for histology by Macenko et al.[28]. Color augmentation (Fig. 1), where the color channels of images are altered at random during training to prevent a model from learning stain characteristics of a specific site have also been utilized in histology deep-learning tasks[29,30]. Most assessments of stain-normalization and augmentation techniques have focused on the performance of models in validation sets, rather than true elimination of the site-specific signature that may lead to model bias[31,32]. Here, we describe the clinical and slide-level variability between sites in TCGA that constitute site-specific digital histology signatures, and methods to ensure robust use of internal and external validation to minimize false-positive findings with deep learning image analysis.

## Results

**Characterization of clinical and digital imaging heterogeneity in TCGA.** Important clinical variables differ across tissue submitting sites across TCGA. It has been recognized previously that outcomes and survival vary across sites for a number of cancers[33], but even more fundamental factors differ depending on submitting organization. We compared the distribution of basic demographics such as age, ancestry, gender, and body weight index and tumor-specific factors such as stage and histologic subtype. Sites were included for comparison if they submitted at least 20 tissue slides. For breast cancer (BRCA TCGA cohort), all demographic characteristics as well as estrogen receptor status ($n = 969$), progesterone receptor status ($n = 966$), HER2 expression ($n = 847$), PAM50 subtype ($n = 914$), TP53 mutational status ($n = 1004$), immune subtype ($n = 1002$), and 3-year progression-free survival ($n = 458$)[34] varied significantly between cohorts, with false discovery rate correction and $P < 0.05$ (Fig. 2). We systematically applied this approach to five other major solid tumor types, and demonstrate that multiple impactful clinical features vary by the site for all tumor subtypes tested—including *ALK* fusion status in squamous cell lung cancer (LUSC TCGA cohort, $n = 155$) and lung adenocarcinoma (LUAD TCGA cohort, $n = 112$) and human papillomavirus (HPV) status in head and neck squamous cell carcinoma (HNSC TCGA cohort, $n = 332$)—all with $P < 0.05$ and significant after FDR correction (Supplementary Table 1 and Supplementary Fig. 1). Of note, given the increasing interest in developing survival models based on pathology, stage varied by the site in all cancer subsets tested, and 3-year progression-free survival (PFS) varied across the site in all cancers, except lung and colorectal adenocarcinoma.

We then applied classical descriptive statistics for image analysis to document the differences in slide image characteristics across site, calculating first-order statistics and second-order Haralick texture features for comparison across sites[35,36]. All first- and second-order statistics demonstrated variance according to tissue submitting site among sites, as measured by ANOVA F-statistic (Fig. 3 and Supplementary Table 2). Similar findings were seen in the analysis of other cancer subtypes (Supplementary Fig. 2). Applying stain-normalization techniques at a slide level for breast cancer improved some first-order characteristics but measures of dissimilarity for all second-order characteristics (as measured by F-statistic) remained greater than that of any first-order characteristics (Fig. 4 and Supplementary Table 2). Of note, the second-order feature angular second moment remained the most dissimilar image characteristic (highest F-statistic) with any form of stain normalization for all cancer types, except lung and head and neck squamous cell carcinoma (Supplementary Table 2 and Supplementary Fig. 2).

**Deep-learning algorithms accurately identify tissue submitting site.** To assess the ability of deep learning to predict tissue submitting sites, we trained a deep-learning convolutional neural network based on Xception architecture to predict site[37]. To

**Batch Effect**

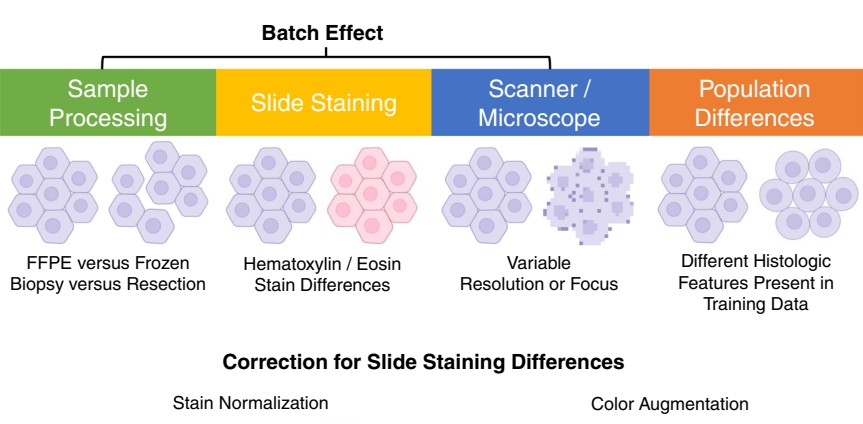

**Correction for Slide Staining Differences**

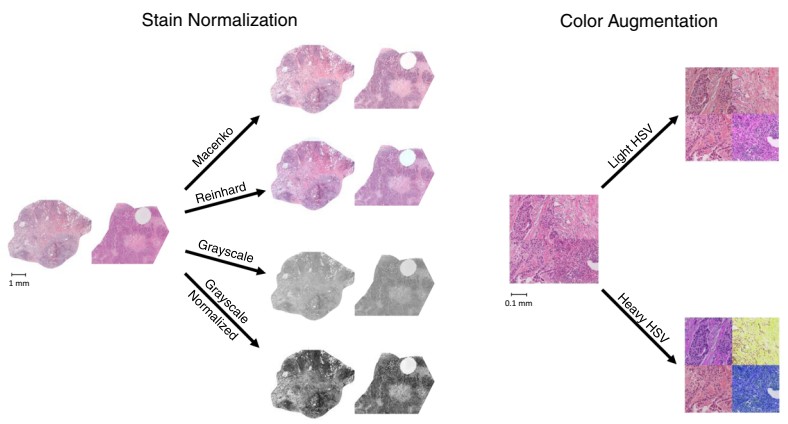

**Fig. 1 Etiologies of site-specific digital histology signatures, and methods for correction.** The submitting institution of a digital histology image can often be readily detected due to a site-specific signature unique to each institution. A number of factors can contribute to site-specific signatures, ranging from true histologic and biologic differences between datasets to nonbiologic artifacts, termed batch effect. The batch effect can originate from every step from the procurement of tissue to digital image creation. Frozen and formalin-fixed specimens will have unique histologic artifacts, the intensity of hematoxylin and eosin exposure can vary between institutions, and the digitization of slides may result in compression artifacts. A variety of methods have been developed to mitigate the impact of stain differences between slides. Stain normalization refers to changes in color characteristics to reduce the effect of staining differences between slides. Augmentation refers to random variations applied to individual tiles during machine learning to prevent overfitting with regards to the varied characteristic.

assess the accuracy of site prediction, we used threefold cross-validation stratified by site (Fig. 5a) and calculated the one-versus rest area under the receiver-operating characteristic (AUROC) curve (Supplementary Table 2). The slide characteristics used by such a model to predict site can be illustrated with a UMAP[38] representation of final layer activations, with representative slide tiles selected for each UMAP coordinate— in this case, demonstrating a hematoxylin-predominant to eosin-predominant color gradient for patients in TCGA-BRCA ($n = 1006$, Fig. 5b). To assess the ability of stain normalization and color augmentation to prevent prediction of site, we repeated this process with normalization or augmentation applied at the tile level for the six examined cancer subtypes (Supplementary Table 3). Site discrimination was highly accurate at baseline, with an average one-versus- (OVR) area under the receiver-operating characteristic curve (AUROC) ranging from 0.998 for clear cell renal cancer (TCGA-KIRC, $n = 508$) to 0.964 for TCGA-LUSC ($n = 463$). For comparison, AUROC for a neural network model trained to predict site from the clinical characteristics described in Supplementary Table 1 achieved an average AUROC of 0.623, ranging from 0.511 in TCGA-LUSC to 0.781 in TCGA-COADREAD (Supplementary Table 4). Stain-normalization techniques modestly decreased the accuracy of site prediction, but site prediction remained highly accurate with an average OVR AUROC of over 0.850 with all normalization techniques for all cancers. For all cancer subtypes tested, the greatest decline in AUROC for site prediction was seen with one of the two forms of grayscale normalization. To further evaluate how stain normalization influences model inference of site, a UMAP and mosaic representation of TCGA-BRCA site prediction after Macenko normalization was generated, which did not demonstrate as clear a color gradient (Supplementary Fig. 4a). The most clearly separable site (A7—Christiana Healthcare) remains the same in both plots—suggesting that either subtle stain-related differences persist, or other components of its unique digital histology signature continue to render this site unique from others.

**An artificial simulation of site-specific digital histology signatures.** As described earlier, there are a variety of putative causes of site-specific signatures in digital histology (Fig. 1) that may contribute to highly accurate detection of the tissue submitting site for a slide. To better describe the relationship between biological factors (such as true differences between populations) and batch effect (i.e., nonbiologic differences between histologic images), we designed a simulation of site-specific signatures using patients from the University of Pittsburgh, the largest contributor to the TCGA-BRCA cohort ($n = 115$ ER-positive, $n = 23$ ER-negative). A single site was chosen as this would theoretically minimize any batch effect due to site-related differences in sample procurement, staining, or digitization. We assessed the ability of deep-learning models to identify 23 random slides from within a cohort of 69 patients while introducing both a biologic cofounder (ER status) and stain-related cofounder—representing two different contributors to a site-specific signature. ER status was chosen as the biologic cofounder as it is highly detectable from

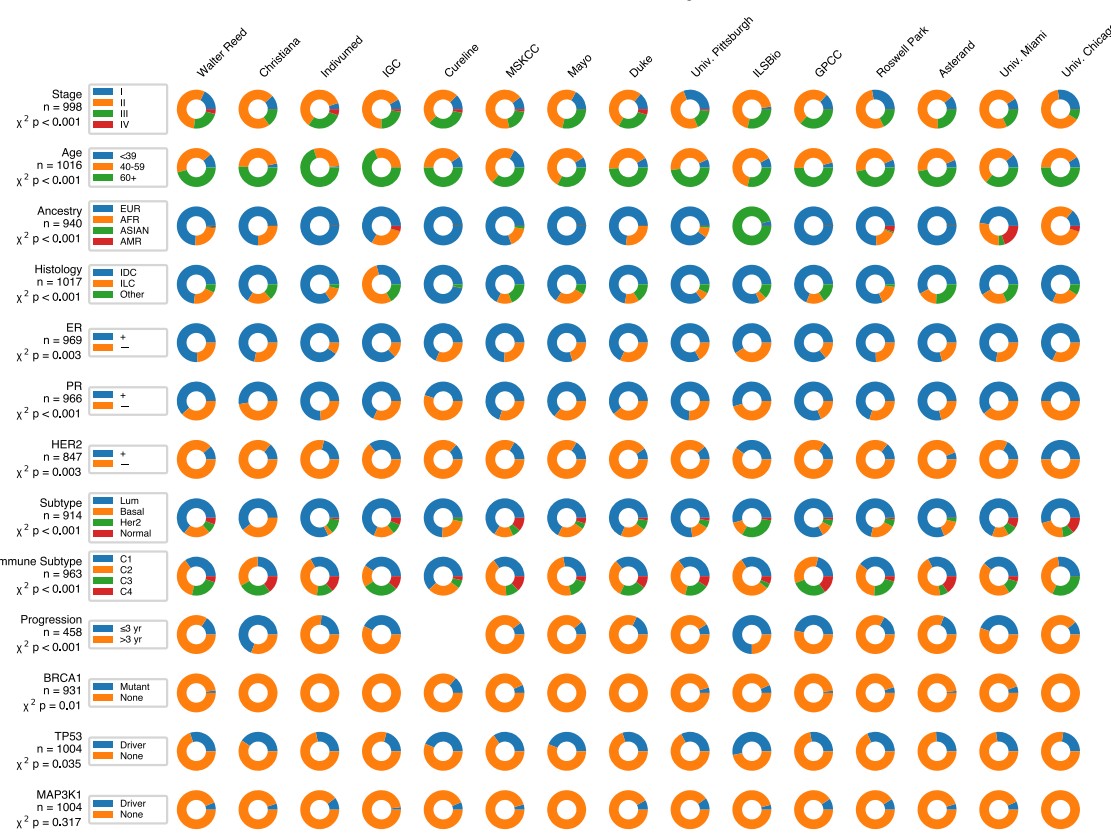

**Fig. 2 Demographics and tumor characteristics of breast cancer across sites with 20 or more slides in TCGA.** Each row represents a demographic, clinical, or tumor characteristic of patients in TCGA-BRCA. The chi-squared test was performed to quantify heterogeneity between sites, with the listed $P$ values corrected for a false discovery rate of 0.05. A number of features display marked heterogeneity—for example, only two sites (ILSBio and Christiana) submitted patients where the majority had disease progression within 3 years. IGC International Genomics Consortium, MSKCC Memorial Sloan Kettering Cancer Center, GPCC Greater Poland Cancer Center, EUR European, AFR African, AMR Native American, IDC invasive ductal carcinoma, ILC invasive lobular carcinoma.

histology, and the University of Pittsburgh dataset has a reasonable number of positive and negative samples. We varied the ER negativity of the 23 target slides from 0 to 100%, whereas the remainder of the slides were maintained as ER-positive (Supplementary Fig. 5 and Supplementary Table 5). Similarly, we applied an artificial staining artifact to 0–100% of the target slides, whereas the remainder of the slides were unaffected. While the accuracy of target feature prediction increased monotonically when the target feature became more strongly ER-negative, this relationship no longer held as the stain artifact was applied to more slides. In addition, stain-normalization techniques did not abrogate the impact of the artificial stain artifact, with a reduction from an AUROC of 1.00 when 100% of target slides had staining artifact, down to a minimum AUROC of 0.934 with grayscale stain normalization. The accuracy at baseline and reduction with grayscale normalization mirrors the ranges of AUROCs seen with site prediction, further suggesting that batch effect, as opposed to biologic subpopulation differences, are the predominant cause of highly accurate site prediction by deep-learning models.

**Preserved-site cross-validation—a quadratic programming solution.** Naturally, if a deep-learning model can distinguish sites based on nonbiologic differences between slide staining patterns and slide acquisition techniques, models designed to predict certain clinical variables could instead learn staining variability or other site-specific features. This is analogous to the Husky versus Wolf problem, where a deep-learning model distinguishes

pictures of these two canines based on the fact that more wolves are pictured in the snow—rather than physical differences between the two animals, leading to a potential lack of external validity[39]. A similar problem can also occur if true biologic subpopulation differences (rather than batch effect) are correlated with the outcome of interest, but only in specific sites. To evaluate the dependence of deep-learning model accuracy on site-specific digital histology signatures, we propose comparing models trained to assess features of interest using two different methods of cross-validation (Fig. 5c). We can correct for biased results by ensuring sites are isolated to a single data fold, or preserved site cross-validation. However, if submitting sites within a dataset are randomly split into equal-sized groups for cross-validation, it is likely that a feature of interest would not be evenly represented among these groups, resulting in biased estimates of accuracy[40]. Optimal stratification for $k$-fold cross-validation while isolating each site to an individual $k$-fold can be achieved using convex optimization/quadratic programming[41]. In other words, an optimization problem can be constructed with the goal of equalizing the proportion of patients with/without a feature of interest across each fold. We applied this method of cross-validation to all outcomes listed in Fig. 2 and Supplementary Table 1. Notably, our method of preserved site cross-validation produces perfect stratification (all subgroups with identical distribution to standard cross-validation) in 55% (32/58) of outcomes tested (Supplementary Table 6). Meaningful imbalances, where the distribution of patients differed from perfect stratification by over 10 for a subgroup in any fold was seen in 12% (7/

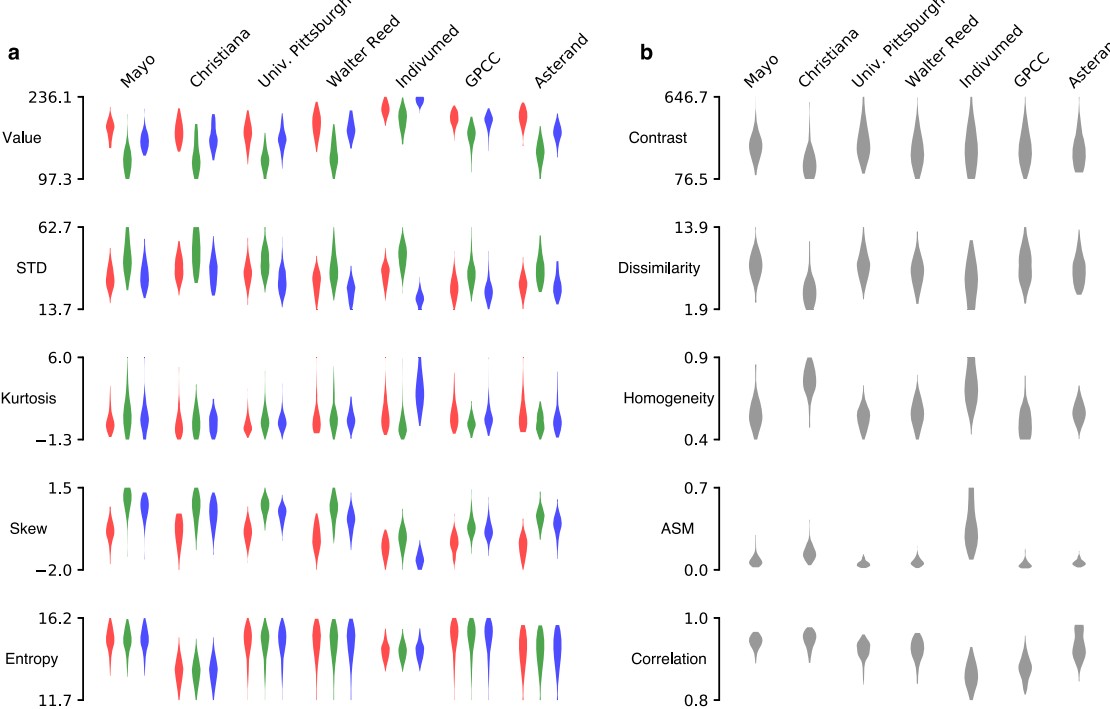

**Fig. 3 Variation of image characteristics in breast cancer digital histology across TCGA.** Sites contributing at least 50 slides are included ($n = 607$ slides, 7 sites), demonstrating that image variation is not solely a function of small sites that infrequently contributed to TCGA. **a** First-order characteristics for red, green, and blue are shown in their respective colors. **b** Haralick second-order textural features also vary by submitting site. STD  standard deviation, ASM  angular second moment, GPCC  Greater Poland Cancer Center.

58) of outcomes. All of these meaningful imbalances occurred in the TCGA-COADREAD dataset, where a smaller number of sites contributed to patients.

**Impact of site-specific digital histology signatures on deep-learning model performance.** To further characterize the influence of site-specific signatures on deep-learning model performance, we trained convolutional neural network models with standard and preserved-site cross-validation to predict the previously described demographic, clinical, and genomic outcomes across six cancer subtypes using the dataset splits as highlighted in Supplementary Table 6. For 58 features evaluated, the average decrease in AUROC between standard and preserved-site cross-validation was 0.069 (range: −0.042 to 0.291). We assessed which models had a significant decline in performance using a one-sided $t$-test, and again repeated this assessment with stain-normalization and augmentation techniques, using an FDR of 0.05 for significance testing. Of the 56 features which were predictable with standard cross-validation, 51 (91.1%) had a decline in AUROC with preserved-site cross-validation, and 20 (35.7%) were no longer significantly detectable (Fig. 6a and Supplementary Tables 7 and 8). A similar proportion of predictable features had a decline in AUROC with other methods of stain normalization/augmentation, ranging 84.6% with grayscale (Fig. 6b) to 89.1% with heavy HSV augmentation. Interestingly, the percentage of features that were no longer accurately detected with preserved-site cross-validation decreased modestly with normalization/augmentation, ranging from 17.5% with Macenko normalization to 26.8% with Reinhard normalization.

Of demographic features, the accuracy of genomic ancestry[42] prediction declined drastically with preserved-site cross-validation in a number of disease subtypes regardless of normalization/augmentation, including TCGA-BRCA ($n = 905$, AUROC 0.798 versus preserved-site AUROC of 0.507, $P < 0.001$), TCGA-

COADREAD ($n = 483$, AUROC 0.883 versus 0.795, $P < 0.001$), and TCGA-LUSC ($n = 422$, AUROC 0.789 versus 0.504, $P < 0.001$). Accuracy of age prediction in the TCGA-COADREAD cohort also declined with preserved-site validation ($n = 541$, AUROC 0.605 versus 0.479, $P < 0.001$), as did stage prediction in both lung cancer cohorts (TCGA-LUSC $n = 474$, AUROC 0.537 versus 0.466, $P < 0.001$; TCGA-LUAD $n = 468$, AUROC 0.599 versus 0.521, $P < 0.001$). As one might expect—these demographic features are often as strongly indicative of disease outcome as pure biologic factors—and outcome prediction demonstrated a significant impact of site-specific signatures in multiple disease cohorts. Performance declined significantly for prediction of 3-year PFS in the TCGA-LUSC ($n = 227$, AUROC 0.589 versus 0.485, $P < 0.001$) and TCGA-HNSC ($n = 272$, AUROC 0.614 versus 0.548) cohorts.

The detection of standard histologic features was less perturbed by preserved-site cross-validation—with no difference in accuracy with the prediction of HER2 status in TCGA-BRCA and of grade in TCGA-HNSC. Other histologic features remained largely unaffected by preserved-site cross-validation—with minimal decreases in AUROC—including prediction of lobular versus ductal histology in TCGA-BRCA, prediction of estrogen, and progesterone receptor status in TCGA-BRCA and prediction of grade in TCGA-KIRC. Prediction of mucinous histology for TCGA-COADREAD, however, did decline with preserved-site cross-validation at baseline ($n = 578$, AUROC 0.788 versus 0.712, $P < 0.001$) and with all forms of normalization/augmentation. Nonetheless, this decline was not dramatic and mucinous histology remained detectable with preserved-site cross-validation.

There has been increasing interest in using deep learning to detect non-intuitive features directly from histology, including our previously described work on detection of genetic driver mutations directly from histology[16]—raising the question of

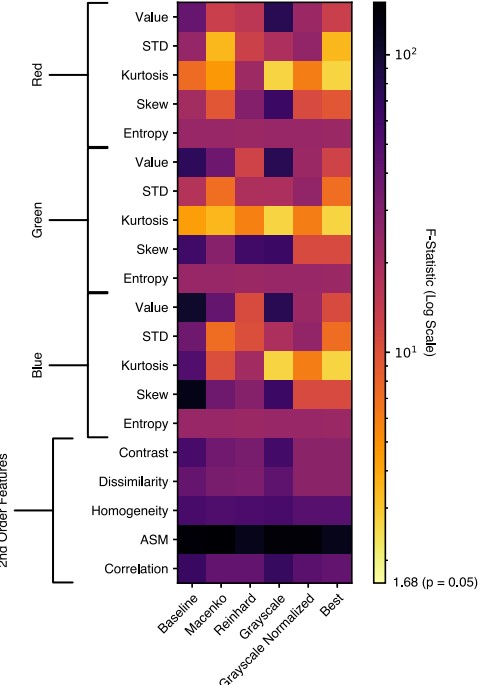

**Fig. 4 ANOVA F-statistic for first- and second-order image characteristics for breast cancer histology in TCGA.** Variance in first- and second-order image characteristics between tissue submitting sites in the breast cancer TCGA dataset ($n = 888$ slides over 14 sites) is assessed with ANOVA. The ANOVA F-statistic is listed for multiple methods of stain normalization, with the lowest F-statistic (least variability) with any method of normalization indicated in the rightmost column. Stain normalization does not completely resolve first-order stain variability (by F-statistic), and minimal impact is seen on second-order Haralick features. STD standard deviation, ASM angular second moment.

whether the accurate prediction of some of these features in TCGA may be due to recognition of site-specific signatures rather than histologic characteristics driven by these mutations. We analyzed a subset of the driver mutations that were accurately predicted in our previous work and found that the majority were unaffected/minimally affected by preserved-site cross-validation, including TP53 and MAP3K1 in TCGA-BRCA, BRAF in TCGA-COADREAD, TP53 in TCGA-HNSC, and STK11 and TP53 in TCGA-LUAD. However, several mutations were no longer accurately detectable, including PIK3R1 in TCGA-LUSC ($n = 458$, AUROC 0.614 versus 0.386, $P < 0.001$), RHOA in TCGA-HNSC ($n = 443$, AUROC 0.733 versus 0.470, $P < 0.001$), and RNF43 in TCGA-COADREAD ($n = 556$, 0.688 versus 0.494, $P < 0.001$). The detection of other genomic features was also dependent on site-specific signatures, including ALK fusion detection in lung cancer (TCGA-LUSC $n = 270$, AUROC 0.678 versus 0.404, $P < 0.001$; TCGA-LUAD $n = 231$, AUROC 0.637 versus 0.417, $P < 0.001$) and immune subtype[34] detection in half of the cancers analyzed.

To further explore why some features exhibit a decline in accuracy, we produced a UMAP and mosaic map of two features in TCGA-BRCA: (1) ancestry, which correlates with site and declined substantially in accuracy (Supplementary Fig. 4b); and (2) BRCA mutational status, which correlates poorly with site and remained detectable with preserved-site cross-validation (Supplementary Fig. 4c). Although the most readily identifiable site (A7, Christiana Healthcare) clusters closely in both, it is not as distinctly separate from other sites in the BRCA UMAP, and a less clear color gradient with BRCA as opposed to ancestry

prediction. This suggests that site-specific histologic patterns weigh less heavily in the decision-making for BRCA mutational status, whereas they may contribute to the prediction of ancestry, resulting in the marked decline with preserved-site cross-validation.

We can further demonstrate that models are weighting the unique histologic pattern of individual sites in making predictions by evaluating model performance within specific sites where patient demographics do not match the overall dataset (Supplementary Fig. 6). We take as an example the slides submitted by the University of Chicago for TCGA-BRCA, the only site where patients of African ancestry comprise the majority of samples. We hypothesized that false-positive predictions of genomic African Ancestry[42] would be significantly higher with standard cross-validation than with preserved-site cross-validation, as models with standard cross-validation may for example learn that the University of Chicago staining pattern is associated with a high rate of African American ancestry. For patients in the validation data folds, false-positive predictions for African ancestry (measured at the tile level, $n = 2206$ tiles, 20 patients, 17 with African ancestry, 3 with European ancestry) are significantly higher for standard cross-validation balanced by ancestry, as compared to preserved-site cross-validation (Fig. 6b and Supplementary Table 9). In other words, standard cross-validation in TCGA inaccurately classifies European patients from a site with predominant African ancestry, as the decision is likely related to nonbiologic site-specific signatures in this multi-site repository.

## Discussion

We have demonstrated that site-specific digital histology signatures exist within TCGA across multiple cancer types, and inadequately controlling for the ease in which deep-learning models detect sites results in biased estimates of accuracy. Although stain normalization can remove some of the perceptible variation and augmentation can mask differences in color, second-order image characteristics are unaffected by these methods, and stain normalization does not resolve the ability of deep-learning models to accurately identify a tissue submitting site. When predicting demographic, clinical, and genetic features with preserved-site validation, a consistent decrease in accuracy is seen despite perfect stratification of features of interest in the majority of cases. The effect size is small for the majority of features and is absent for most features with a clear histologic basis such as tumor histologic subtype and grade. Conversely, we demonstrated that prediction of other clinically relevant features such as progression-free survival for squamous lung cancer and head and neck cancer, as well as genomic features such as certain driver mutations, ALK fusion status, and immune gene expression for certain cancers, are significantly driven by site-specific signatures—despite any form of normalization/augmentation.

Demographic features have a less straightforward histologic basis, but it is not unreasonable to expect that some can be detected from histology. For example, young age is correlated with high-grade tumors and older age associated with lobular histology in breast cancer[43]. A clear biologic link between ethnicity and histology has been demonstrated in breast cancer—with higher tumor grade, more frequent triple-negative receptor status, and recurrent genetic differences in genome-wide association studies characterizing African American breast cancer[44–47]. However, we have demonstrated that deep-learning models trained on multisite repositories such as TCGA may base predictions on the histologic signatures of submitting sites, rather than intrinsic tumor biology, when these site-specific signatures are correlated with the outcome of interest. Demographic features

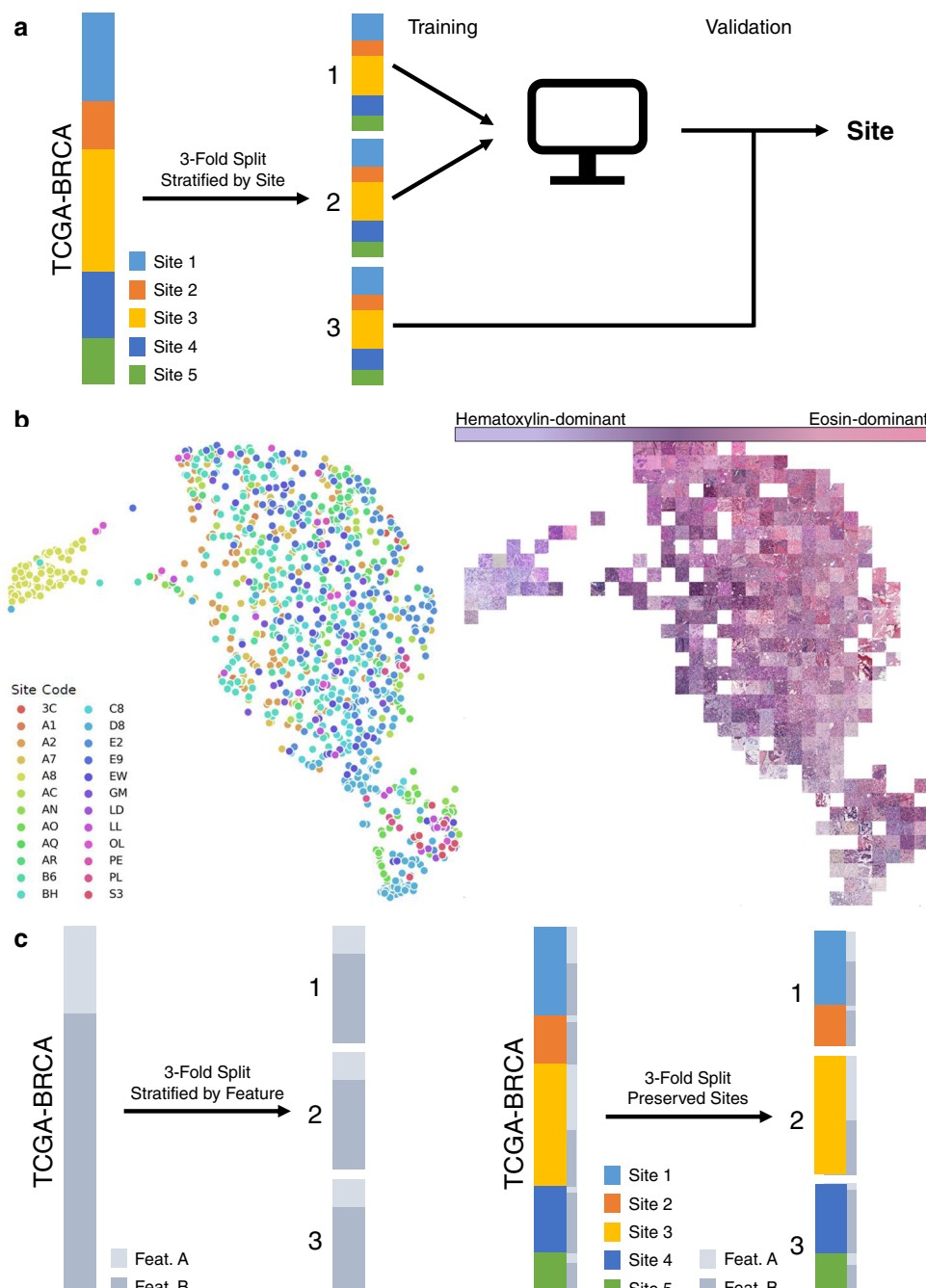

**Fig. 5 Model development for the site and feature prediction for patients in TCGA. a** To predict tissue submitting site, data is split into threefolds, with each site represented equally in all folds. Cross-validation is then performed, where a model is trained on two of the datasets and performance is assessed on the third dataset. This process is repeated threefold for an averaged performance metric. **b** UMAP representation of final activation weight vector of the model trained to recognize submitting site in TCGA-BRCA ($n = 1006$ slides). Each point on the left figure represents the centroid tile from a single slide. The nearest tile to each UMAP coordinate is visualized on the right, demonstrating a clear gradient from tiles that demonstrate predominant hematoxylin staining to those demonstrating predominant eosin. **c** We assess the impact of including slides from a tissue submitting site within both the training and validation sets on the prediction of a variety of clinical, genomic, and demographic features, using two methods of generating folds for cross-validation. First, we split the data into threefolds, stratifying by the feature of interest, irrespective of site. For a comparator, we split the data into threefolds where each site is isolated into a single fold, with the secondary objective of equalizing the ratio of features in each fold.

such as genomic ancestry, which varies greatly from site to site due to differences in catchment areas of hospitals, may be particularly susceptible to such bias. This is evidenced by the fact that ancestry is predictable in TCGA-BRCA with standard but not preserved-site cross-validation, and predictive accuracy for ancestry declined significantly with preserved-site cross-validation for most cancer subtypes. This poses a challenging ethical

dilemma for the implementation of deep-learning histology models. It has been well documented that women of African ancestry with breast cancer have a poorer prognosis that is not completely accounted for by stage and receptor subtype[48,49]. Contributing factors may include delays in treatment initiation and inadequate intensity of therapy[50], and more research is needed to disentangle the biologic and nonbiologic factors

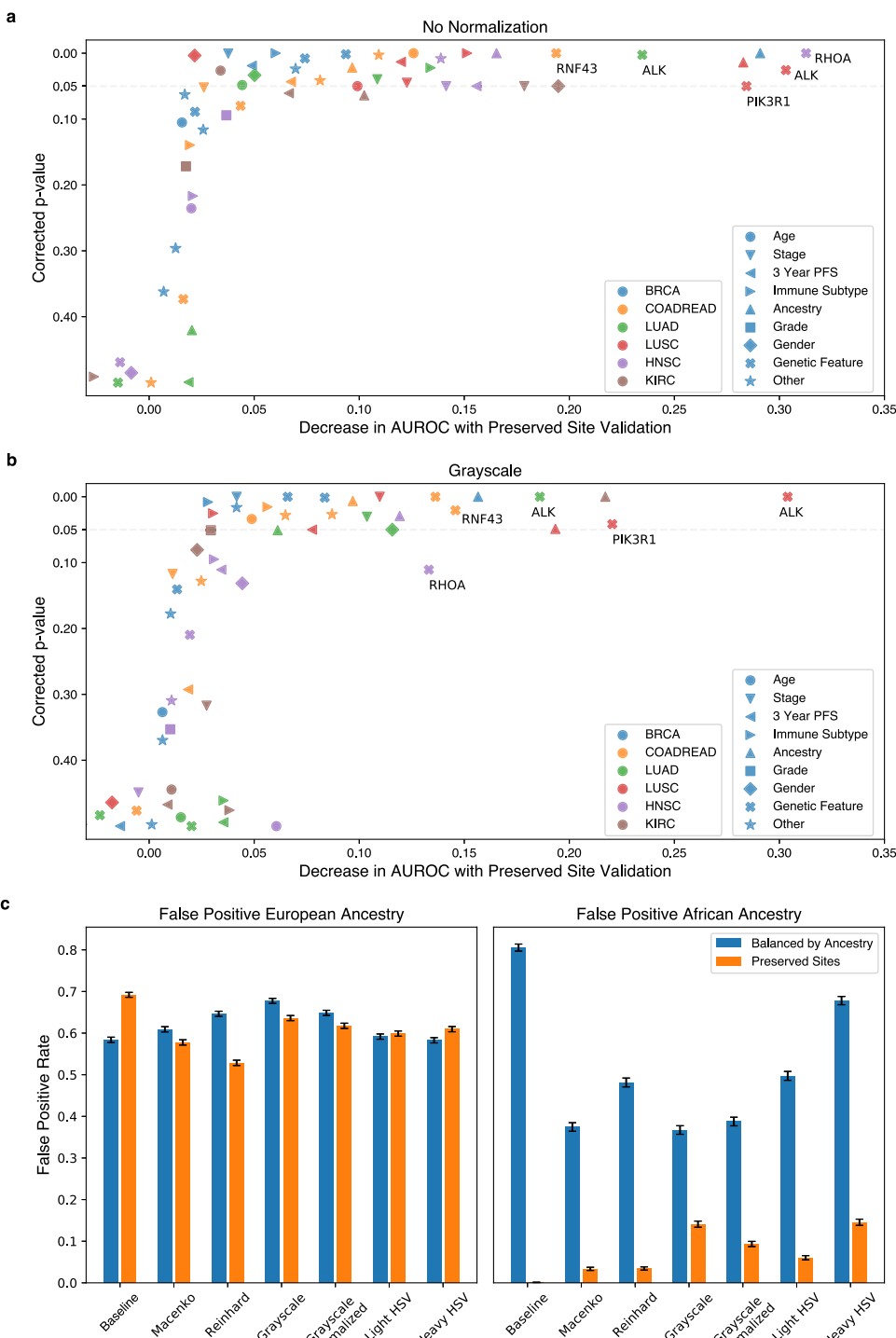

**Fig. 6 Impact of site-specific digital histology signatures on deep-learning model accuracy and bias. a** The distribution of the average difference in AUROC with standard and preserved-site cross-validation for various clinical, genomic, and demographic features ($n = 58$ features) for six cancer subtypes in TCGA is shown (pictured graphic for baseline models without normalization/augmentation). The decrease in AUROC is statistically significant for a number of features (one-sided t-test, illustrated on y axis with false discovery correction, as described in Supplementary Table 7) for a subset of features. Jitter is added to ease visualization, although significance/insignificance of individual findings is preserved. **b** The same graph is provided for grayscale stain adjustment (for which the smallest changes in AUROC were seen). **c** False-positive prediction of European ancestry and African ancestry for patients within the University of Chicago dataset (measured at the tile level, $n = 2206$ tiles from 20 patients, 17 with African ancestry, 3 with European ancestry) for models trained with standard and preserved-site cross-validation. The presented bars illustrate the proportion (e.g., the number of tiles falsely predicted to be European divided by the total number of tiles predicted to be European), with error bars signifying the estimated standard deviation of the proportions. PFS progression-free survival, HSV hue saturation value.

contributing to disparities in prognosis. As deep-learning models are able to infer patient ancestry from site-specific signatures, models must be carefully implemented in an equitable fashion to avoid recapitulating the pre-existing inequities in cancer care[51]. Further study within single-site repositories, or repositories where tissue is stained and digitized at a single center, may promote more accurate modeling of demographic factors with deep learning.

When developing predictive histologic models for a large number of features, external validation of every finding can be impractical/infeasible. Furthermore, adequate external validation datasets may not be readily available for rare cancer subtypes. As such, multiple studies have trained and validated models using TCGA with no external validation or only partial validation in a subset of cancer types. Such studies include genetic mutation prediction in multiple cancer types[16,17,25], prediction of grade in clear cell renal cancer[52], prediction of breast cancer molecular subtype[53], the prediction of gene expression[13], or correlation of histology and outcome[25,54]. Survival outcomes are particularly challenging to develop rigorous models for using histology from TCGA, and model performance may be falsely elevated not only by the disparate outcomes across sites, but also the site-level differences in critical factors relevant to survival such as stage and age. Studies demonstrating histologic discrimination of survival and recurrence in glioblastoma[52,54], renal cell cancer[52], and lung cancer[55] patients from TCGA which lack external validation cohorts may therefore have biased estimates of the outcome. Prediction of survival may also suffer from this bias[56] even when correcting for age, stage, and sex, as other factors that vary by the site also contribute to the outcome, ranging from the ethnicity of enrollees to the treatment available at academic vs community centers. Given that traditional image and textural characteristics vary between sites in TCGA, it is likely that non-deep-learning prognostic studies that predict outcome from traditional image analysis features may suffer from a similar bias[57]. Although prognostic models without external validation must be carefully scrutinized, a number of studies have shown that deep-learning prognostic models can maintain strong performance in external datasets for cancers such as colorectal cancer[58,59] and mesothelioma[60]. Of course, a number of models initially tested in TCGA without preserved-site cross-validation have maintained accurate prediction in external validation cohorts, such as prediction of microsatellite instability or BRAF mutations in colon cancer[11,16]. Notably, in our study, preserved-site cross-validation demonstrated that both BRAF status and MSI status remained detectable without substantial decline with most forms of normalization. However, models developed to predict several other driver mutations suffered significant declines in performance/ were no longer detectable. Similarly, in a study by Fu et al., 0–32% of genetic alterations predictable in TCGA-BRCA were no longer detectable in two external cohorts[25]. It must be noted that the prevalence of some of the genomic alterations evaluated in this study was rare, and thus they may be more susceptible to changes in predictive accuracy just due to random chance rather than from site-specific digital histology signatures. Nonetheless, preserved-site cross-validation may show promise as a tool to identify which features are unlikely to survive the test of external validation prior to extensive additional resource commitment.

We recommend a series of best practices for deep-learning studies on histology using TCGA or other combined datasets of multiple hospital sites. First, the variation of outcomes of interest should be reported across included sites. This will allow an assessment of the potential impact that site-specific signatures can have on accuracy. In addition, knowledge about the distribution of outcomes on the training and testing sites can allow for accurate assessment of model performance, as AUROC is an uninformative marker for heavily imbalanced datasets, where the precision-recall curve can be more informative[61]. Even if performance stands the test of external validation, models may retain the biases learned from institutional staining patterns. Thus, if outcomes of interest vary heavily across sites, further prospective validation at individual institutions may be necessary before implementation.

If variation of outcomes is seen across sites within a multisite repository, models should not be trained and assessed for accuracy on patients from the same contributing site. As we have demonstrated, including a site within both the validation and training datasets results in biased estimates of accuracy. The tried and true gold standard for any artificial intelligence endeavor is external validation, which also ensures that not only site level but dataset level digital histology signatures are not driving model performance[62]. However, adequate external validation datasets are not frequently available, and it is important to accurately assess the promise of models at an early stage before significant time is spent in further research and investigation. We propose using convex optimization/quadratic programming as demonstrated in this study to identify the split of sites to allow optimal stratification of features of interest. This can also be applied to linear features by stratifying the feature of interest into meaningful subgroups or quartiles prior to optimization.

Finally, stain-normalization and color augmentation techniques should still be used to improve model accuracy in external validation and implementation. Although normalization and augmentation do not prevent models from learning site-specific characteristics, several studies have reported greater validation accuracies with the use of such techniques[31,32]. It is likely that these techniques eliminate some but not all of the reliance that deep-learning models have on staining differences; by making the differences in slide characteristics more subtle, models may be more likely to pick up on biologically relevant factors. In our study, Macenko stain normalization maintained the greatest proportion of features that remained detectable with preserved-site cross-validation. However, the best method of augmentation/ normalization to eliminate these biases varies by dataset/feature of interest. Forms of grayscale normalization may better eliminate stain and site detection, but likely discards some relevant biologic information and may impact predictive accuracy[63]. Similarly, although attempting to normalize second-order image characteristics derived from the gray level co-occurrence matrix may render sites more indistinguishable, such characteristics are closely associated with intrinsic tumor biology and must likely be preserved for deep-learning applications[64].

Our findings are not without limitations. In this work, we present a comprehensive description of pixel-level characteristics across sites in TCGA using classical image analysis techniques, however other factors likely contribute to the detectable differences between sites. It is likely that other higher-order image characteristics contribute to the site-level differences, such as Gabor, wavelet packet, and multiwavelet features[24]. However, extensive characterization of all described textural characteristics is not necessary to demonstrate the presence of site-specific digital histology signatures and the impact this has on model performance.

Our method of generating preserved-site cross folds successfully stratified patients by outcomes of interest for the majority of features examined, but there were some notable outliers in the TCGA-COADREAD dataset. For example, stratification of mucinous histology in TCGA-COADREAD was far from perfect and could lead to the slight decline in predictive accuracy for mucinous histology seen with preserved-site cross-validation. Other features such as microsatellite instability (MSI) were also poorly stratified—in the case of MSI status, one validation fold

contained over three-fourths of available patients because one organization contributed the majority of samples where MSI status was known. However, for MSI status, poor stratification did not significantly affect performance when tested with preserved-site cross-validation—consistent with the fact that MSI status has a well-proven histologic basis[10]. This limitation does not apply for the majority of features evaluated, and preserved-site cross-validation can likely be applied to most multisite histology repositories.

Multiple methods for assessment of statistical significance have been proposed for AUROC analysis, including seminal work by DeLong et al.[65]. However, application to the aggregate of predictions using DeLong's method fails to capture the variance in predictive accuracy seen when training with different subsets of data, and also is not extensible to multicategorical models such as those we used for submitting site and stage prediction. Bootstrapping as per Hanley and McNeil[66] is also highly utilized, but in our preliminary analyses, we had planned to assess model performance without bootstrapping. As the number of features we planned on analyzing grew, we updated our analytic plan to include bootstrapping as described to allow for reasonable estimates of significance with false discovery correction, as well as to mirror the methods of our group's prior work in genomic feature detection, allowing for better comparison to these results[16].

Our study focuses on correction and analysis of slide stain differences, which is just one component of potential contributors to the site-specific signatures seen in TCGA (Fig. 1). It is likely that some of the declines in performance seen with preserved-site cross-validation could also be due to differences in specimen processing, slide scanning, or subpopulation differences between enrolling sites. For example, other studies describe that JPEG quality had a strong confounding effect on classification tasks in TCGA[25]. We attempted to minimize the influence of resolution on our findings by sampling slides in our deep-learning models at a fixed pixel to μm ratio, but we did not directly assess the ability of deep-learning models to detect compression. Several of our findings support slide stain differences as a primary etiology of site-specific signatures in TCGA. First, a UMAP of final layer activations for site prediction as well as other highly affected features in TCGA-BRCA highlights that an azurophilic to eosinophilic gradient (Fig. 5b). This suggests that stain variation is one of the most important distinguishing elements used in the prediction of these features, although there may be confounding between staining pattern and JPEG compression artifact. Even basic first-order imaging characteristics such as average red, green, and blue values vary significantly between sites with all methods of stain normalization (Fig. 4), suggesting that stain differences may still play a role in differentiation between sites. Nonetheless, second-order image characteristics vary more than these first-order characteristics—and further study of the impact of staining, choice of slide scanner, and method of sample acquisition on image characteristics can further elucidate the drivers of these differences. When varying both subpopulation differences (ER status) and slide staining in a set of target slides, the influence of slide staining abnormalities clearly predominates and reduces the impact of biologic differences on accuracy (Supplementary Fig. 5). Thus, when significant slide staining differences are present (as seen in Fig. 5b), the influence of biologic differences is likely minimal. Furthermore, the pattern of decline of the artificial stain shift mirrors what was seen with stain normalization for site detection (Supplementary Table 3), suggesting that the use of stain normalization does not eliminate the effect of stain differences. Although the etiology of the decline in performance with preserved-site validation is debatable, preserved-site cross-validation may provide valuable insight into performance on external datasets when site-specific staining,

scanning, specimen processing, or subpopulation differences are present. However, it must be noted that preserved-site cross-validation has the potential to negate true biologic associations between histology and features of interest if these associations are only present in a single site. We have also only chosen a subset of proposed stain correction methods, but there have been other approaches that may further reduce the intrasite variability in staining. An unsupervised learning approach to normalizing stains has been proposed, but did not outperform augmentation in test datasets[31]. Adversarial networks may also allow for models to avoid learning undesirable characteristics of datasets[67].

In summary, we have demonstrated that digital histology in TCGA carries a multifactorial site-specific signature that is characteristic of the tissue submitting site. This signature can be easily identified by deep-learning models and can lead to an overestimation of model accuracy if multiple sites are included in both the training and validation datasets. We have demonstrated that this site-specific signature can lead to the appearance of accurate prediction of clinical findings ranging from progression-free survival, gene expression, genetic mutations, and ancestry with standard cross-validation. Care should be taken to describe the distribution of outcomes of interest across sites, and if significant, a submitting site should be isolated to either the cohort used for training or for testing a model. A quadratic programming approach can maintain optimal stratification while still isolating submitting sites to either training or validation datasets.

## Methods

**Patient cohorts.** Patient data and whole-slide images were selected from six of the tumor types from TCGA with the highest number of slides available to better identify site-specific digital histology signatures. Tumor types included breast (BRCA)[68], colorectal (COAD and READ – with data combined for sites enrolling to both cohorts)[69], lung squamous cell carcinoma (LUSC)[70], lung adenocarcinoma (LUAD)[71], renal clear cell (KIRC)[72], and head and neck squamous cell carcinoma (HNSC)[73]. Slides and associated clinical data were accessed through the Genomic Data Commons Portal (https://portal.gdc.cancer.gov/). Ancestry was determined using genomic ancestry calls provided by Carrot-Zhang and colleagues, with computation as described in their work[42]. Immune subtypes were used from the work published by Thorsson et al.[34]. Informed consent was obtained for all participants in TCGA, and ethics oversight is described at https://www.cancer.gov/about-nci/organization/ccg/research/structural-genomics/tcga/history/policies.

**Image processing and deep-learning model.** Scanned whole-slide images of hematoxylin and eosin-stained tissue were acquired in SVS format from TCGA. Each slide was reviewed by a pathologist for manual annotation of the area of the tumor using QuPath version 0.12, to ensure ink or other non-cancer artifacts did not influence slide-level statistics[32]. For analysis of first-order and second-order image characteristics, slides were downsampled to 5 microns per pixel or approximately ×2 magnification. For deep-learning applications, the tumor region of interest is tessellated into 299 × 299 pixel tiles for evaluation, each representing a 302 × 302 μm area of histology, effectively generating an optical magnification of ×10. A more in-depth description of our preprocessing methodology is publically available (https://doi.org/10.5281/zenodo.3694994). An average of 1% of slides was excluded for quality issues as described in Supplemental Table 6. Convolutional neural network models are written in Python 3.8 with TensorFlow 2.3.0, using the Xception model architecutre[37] pre-trained on the ImageNet database[74]. The final fully connected layer of Xception is replaced by a single fully connected hidden layer with width 500, followed by a softmax layer for prediction. This architecture is analogous to what is used in other large pan-cancer studies of TCGA to allow comparison of our findings to such studies[16,19,25]. Models were trained over 3 epochs of data, using the Adam optimizer[75], with a learning rate of $10^{-6}$, a batch size of 32, sparse categorical cross-entropy loss, and no L2 regularization or dropout. For the prediction of cancer sites using clinical tumor characteristics alone, a similarly structured neural network was used with a single hidden layer with a width of 500.

Each tile is assigned a label associated with the outcome of interest. Tile libraries were also balanced by category to eliminate bias, with downsampling such that the number of tiles for each target category was equivalent. Stain normalization and augmentation are applied to individual tiles at the time of training and assessment. Macenko and Reinhard normalization is applied as previously described[27,28] using a publically available implementation[76], grayscale refers to direct slide conversion to grayscale, and "grayscale normalized" refers to conversion to grayscale with histogram equalization[77]. Both light and heavy levels of hue saturation value (HSV) augmentation was applied, with light augmentation multiplying each of these three

channels by a scalar from 0.9 to 1.1, and heavy augmentation multiplying the hue and saturation channels by a random scalar from 0.7 to 1.3. In addition, further augmentation through random tile rotation is performed, and further normalization ensures inputs have a mean of zero and a variance of one. Models are trained with threefold cross-validation, learning from two splits of the data and then evaluated on the third split (Fig. 5). Deep-learning model training and evaluation was performed on 16 deep-learning-specific NVidia Tesla V100s graphical processing unit (GPU) nodes within a HIPAA-compliant environment.

**Statistics and reproducibility**. To quantify differences between categorical clinical features across sites, a Chi-squared test is used for sites submitting over 20 slides, with significance determined using a false discovery rate (FDR) of 0.05 with the Benjamini–Hochberg method applied individually to each cancer subtype (Supplementary Table 1). The 20 slide cutoff was chosen for these descriptive analyses to prevent variance metrics from being driven by sites submitting small numbers of slides that may be skewed due to sampling errors. The number of patients analyzed and degrees of freedom for each analysis is described in Supplementary Table 1. The variability in site-level image characteristics is quantified in this same set of sites, using the ANOVA F-statistic to measure variation for each individual image characteristic, with degrees of freedom equal to one less than the number of included sites, with the same FDR (Supplementary Table 2). First-order statistics are calculated from individual red, green, and blue pixel values across images, and include mean, standard deviation, skewness, kurtosis, and entropy, the latter being calculated as follows:

$$\text{Skewness} = \frac{m_3}{m_2^{3/2}} \tag{1}$$

$$\text{Kurtosis} = \frac{m_4}{m_2^2} \tag{2}$$

$$\text{Where } m_i = \frac{1}{N} \sum_{(n=1)}^{N} (x_n - \bar{x})^i \tag{3}$$

$$\text{Entropy} = -\sum_{n=1}^{N} x_n * \log(x_n)$$

Second-order Haralick features[36] were calculated from the gray level co-occurrence matrix **P**:

$$\text{Contrast} = \sum_{i,j=0}^{255} \mathbf{P}_{i,j}(i - j)^2 \tag{4}$$

$$\text{Dissimilarity} = \sum_{i,j=0}^{255} \mathbf{P}_{i,j}|i - j| \tag{5}$$

$$\text{Homogeneity} = \sum_{i,j=0}^{255} \frac{\mathbf{P}_{i,j}}{1 + (i - j)^2} \tag{6}$$

$$\text{Angular smooth momentum} = \sum_{i,j=0}^{255} \mathbf{P}_{i,j}^2 \tag{7}$$

$$\text{Correlation} = \sum_{i,j=0}^{255} \mathbf{P}_{i,j} \frac{(i - \mu_i)(j - \mu_j)}{\sqrt{\sigma_i^2 x \sigma_j^2}} \tag{8}$$

Similar values for calculated features were seen for angles of 0°, 45°, 90°, and 135° so reported values for second-order features are averaged across these four angles. Second-order image characteristics were calculated using the python scikit-image library, version 0.18.0[78].

Deep-learning model predictions are assessed with the area under the ROC curve (AUROC), averaged over the threefolds generated for cross-validation. Confidence intervals and statistical testing were computed using a ×10 bootstrapped experiment. For multicategorical models (such as the prediction of the tissue submitting site of a slide, or prediction of Stage I vs II vs III disease), the reported AUROC values are the one-versus rest AUROC, calculated using the Scikit-learn library, version 0.23.2.

For the prediction of tissue submitting sites, deep-learning models were trained using the aforementioned architecture. Comparisons between average OVR AUROC for prediction of site with different methods of stain normalization was performed with a two-sided paired t-test with two degrees of freedom and an FDR of 0.05, and comparisons to assess if average AUROC was greater than random chance (AUROC 0.50) were performed with a one-sided t-test with an FDR of 0.05 and two degrees of freedom (Supplementary Table 3).

Deep-learning models were also trained in a series of artificial experiments to predict a simulated feature of interest within the University of Pittsburgh dataset ($n = 115$ ER-positive, $n = 23$ ER-negative). Models were trained to detect 23 patients with a varied percentage of ER positivity (0–100%) and varied percentage of stain alteration (0–100%, as per Supplementary Fig. 5 and Supplementary Table 5). The stain alteration consisted of a 0–5% increase in hue, saturation, and value. These patients were combined with 46 ER-positive patients with no stain

alterations, and accuracy of prediction of the feature of interest was assessed with average AUROC with threefold cross-validation.

Accuracy for prediction of clinical variables is reported with standard cross-validation—stratifying by site, and with preserved-site cross-validation—where each site is isolated to a single fold, and secondarily stratifying by the site. In other words, for standard cross-validation, all sites are merged into a single dataset, and folds are created without respect for site, such that classes are balanced and three equal folds are produced. Conversely, for preserved-site cross-validation, the dataset is divided into several folds, such that patients from a single site are all contained within the same fold. The split of sites is also selected to ensure the distribution of patients in each fold with respect to the outcome of interest is reflective of the larger population. To calculate k-folds for preserved-site cross-validation, we define the following convex optimization problem. If $m_{s,c}$ is a binary variable indicating if site $s$ is a member of fold $c$, and $n_{s,f}$ is an integer indicating the number of samples from the site in the categorical feature class $f$, then we seek to minimize the mean squared error of divergence from perfect stratification:

$$\text{Error} = \sum_{\substack{f \in \text{Features,} \\ c \in \text{Crossfolds}}} \left( \sum_{s \in \text{Sites}} m_{s,c} \cdot n_{s,f} - \sum_{s \in \text{Sites}} \frac{n_{s,f}}{k} \right)^2 \tag{9}$$

With the constraints that for all sites $s$:

$$\sum_{c \in \text{Crossfolds}} m_{s,c} = 1 \tag{10}$$

We used CPLEX v12.10, IBM to solve the optimal solution of Eqs. (9) and (10)[79]. Our code used for fold generation for preserved-site cross-validation is available from https://github.com/fmhoward/PreservedSiteCV[80].

We assess the impact of site-specific signatures on model accuracy across 58 features for the 6 aforementioned cancer types using both standard and preserved-site validation—the cross-validation folds and sample sizes used in this assessment are listed in Supplementary Table 6. To assess if site-specific signatures reduce the performance of models, we used a one-sided t-test with four degrees of freedom to compare standard and preserved-site cross-validated AUROCs. One-sided t-test was chosen given the distribution in Fig. 6a which suggests that preserved-site validation does not improve model accuracy. To assess if a model significantly predicts a feature of interest, we assess if reported AUROC values for the two methods of cross-validation are greater than random chance (0.50) using a one-sided t-test. For example, in some cases, a feature may be accurately predicted with no stain normalization, but is no longer accurately predicted with grayscale normalization. Thus, there may be a decline in performance with preserved-site cross-validation with no stain normalization, but no decline in performance with grayscale normalization (as neither model can make any accurate predictions in grayscale). Both comparisons are performed for all methods of stain normalization, with an FDR of 0.05 for each feature analyzed (Supplementary Tables 7 and 8). Comparisons between the false-positive rates for African ancestry in TCGA-BRCA were performed using a chi-squared test at a tile level with one degree of freedom and an FDR of 0.05 (Supplementary Table 9).

To ensure reproducibility, computer code used to support the main findings of this work was run in duplicate with equivalent results, including the subdivision of sites into groups for cross-validation using the provided software.

**Reporting summary**. Further information on research design is available in the Nature Research Reporting Summary linked to this article.

## Data availability

Data from TCGA including digital histology and the clinical and genetic annotations used are available from https://portal.gdc.cancer.gov/ and https://cbioportal.org. Annotations for immune subtypes are available from the published work of Thorsson et al.[34] (https://doi.org/10.1016/j.immuni.2018.03.023), and annotations for genomic ancestry were directly obtained from the work of Carrot-Zhang et al.[42] (https://doi.org/10.1016/j.ccell.2020.04.012). Annotations for driver mutations are available from https://github.com/jnkather/DeepHistology. All other results in support of this manuscript are available from the corresponding author upon reasonable request. Source data are provided with this paper.

## Code availability

Our code used for fold generation for cross-validation is available from https://github.com/fmhoward/PreservedSiteCV[80].

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

## Acknowledgements

A.T.P received support from the NIH/NIDCR (K08-DE026500), the NCI (U01-CA243075), the Adenoid Cystic Carcinoma Research Foundation, the Cancer Research Foundation, and the American Cancer Society. D.H., R.N., and O.I.O received support from the NIH/NCI (1P20-CA233307). Figure 1 and Supplementary Figure 5 were created in part with BioRender.com.

## Author contributions

F.M.H. and A.T.P. were responsible for concept proposal and study design. F.M.H., J.D., S.K., and J.N.K. performed essential programming work. J.S., H.C., L.H., and N.C. performed manual oversight and quality control for digital pathology, along with segmentation of tumor. F.M.H., J.D., S.K., D.H., R.N., O.I.O., J.N.K., B.G., and A.T.P. contributed to data interpretation and statistical approaches. All authors contributed to the data analysis and writing of the manuscript.

## Competing interests

R.N. reports relationships with Aduro, Cardinal Health, Clovis, Fujifilm, G1 Therapeutics, Genentech, Immunomedics/Gilead, Ionis, iTeos, MacroGenics, Merck, Oncosec, Pfizer, Seattle Genetics, serves on the data safety monitoring board for G1 Therapeutics, and receives research funding from Arvinas, AstraZeneca, Celgene, Corcept Therapeutics, Genentech/Roche, Immunomedics/Gilead, Merck, OBI Pharma, Odonate Therapeutics, OncoSec, Pfizer, Seattle Genetics, Taiho. O.I.O. reports relationships with CancerIQ, Tempus, and 54gene, and speaks as an Advocate for Susan G Komen and the American Cancer Society. J.N.K. reports a relationship with Owkin. A.T.P. reports a relationship with New Rhein and serves on the advisory board for Prelude Therapeutics.
