## [Peer Review File · Nature Communications]

Reviewers' Comments:

Reviewer #1:

Remarks to the Author:

The Impact of Digital Histopathology Batch Effect on Deep Learning Model Accuracy and Bias

Overall comments:

I like the premise. In fact, I had a discussion with the editor of Nature MI on the importance of batch effects on ML performance. The authors touch on a very important topic, one that has clear implications on how AI/ML can be adapted for use on clinical data.

However, I feel that the authors do not have a good sense of what batch effects are --- they seem to confuse clinical and subpopulation effects as batch effects (which are technical biases).

They also attribute the performance of a feature as "batch correlated" if a feature provides good performance in random data as opposed to the preserved site cross validation --- this is not necessarily due to the influence of a batch effect.

I think there is value in carefully and repeatedly stratifying sample to carefully test for the confounding effects of some variable. But the weakness of such an approach is that the confounding variable must be known and described in the meta-data in order for it to be evaluated and tested. It is also messy and tedious to perform.

I also think there is limited value in saying generally about idiosyncratic batch effect, which is confounded with other biological/clinical subfactors, and then saying that stratification can help minimize such effects. I think it is more useful if a more controlled experiment can be used to illustrate the impact of the batch effect on ML given some assumptions about its nature and distributions.

I have the following specific comments:

"Furthermore, ethnicity can also be inferred from histology due to site-level batch effect, which must be accounted for to ensure equitable application of DL."

May not be batch effect. Unless the batch factor is also confounded with ethnicity information.

"Naturally, if a deep learning model can distinguish sites based on non-biologic differences between slide staining patterns and slide acquisition techniques, models will learn the clinical variability between samples at different sites."

This happens regardless of whether the non-biologic difference exists or not.

"First, stratifying by feature of interest and ignoring site, a number of features may be predicted due to batch effect between sites. We can correct for biased results by ensuring sites are isolated to single data fold, or preserved site cross validation."

This expression is convoluted. What I think it sounds like to me is that you are comparing two groups where no biologic differences exist except the batch factor. In other words, you are looking for batch-correlated variables.

But what happens if a feature is both class and batch confounded? Do you reject or drop it anyway?

"However, it is still necessary to stratify by the variable of interest, as stratification improves the variance and bias of estimates of accuracy³⁸"

This follow up sentence is weird. You already said to stratify by feature of interest. And now you say it is still necessary to do this?

I also think direct-copy statements like "stratification improves variance and bias of estimates of accuracy" is not helpful. You mean to say it reduces variance and bias.

"It is not possible to perfectly stratify by race, but we can select partitions of sites as close as possible for k-fold cross-validation using convex optimization / quadratic programming³⁹"

I think having multiple partitions based on single variable splits is not very useful.

"Essentially, we optimize an error represented by the deviation of a validation fold from perfect stratification, where the percentage of patients in each subgroup is equal to the average percent over the dataset."

This is a very convoluted way of simply saying --- the proportion of patients in each fold is kept constant.

So you are trying to adjust for situations where your fold is not representative of the label proportions of the original data.

But you also said that for certain features, it is not possible to get a good stratification.

And also, how do you know that all meaningful and important features that biases your model are recorded?

Also, no mention of the model type you use? Maybe the outcome is an artifact of the model or hyperparameter settings?

"We applied this method of stratification to all outcomes listed in Figure 2 and Supplemental Table 1."

I am confused here. First you say you are optimizing a method that accounts for deviation from sample proportions.

Now you are saying that your method is a stratification approach.

"For 49 features evaluated, an average decrease in AUROC of 0.064 (range, -0.012 to 0.309) was seen, with a number of models demonstrating a meaningful decrease in performance attributable to batch effect related bias (Figure 6a, Supplemental Table 5)."

I think whenever a drop in performance is observed, it may not necessarily be attributed to just "batch".

"...for TCGA-LUSC declined significantly with preserved site cross validation regardless of stain normalization / augmentation."

So this means stain norm and augment did not work. But made things worse.

But I am also confused, you said you assessed which models had the significant decline. But here, you only allude to one particular example.

"...identification of a feature of interest from histology is ostensibly solely due to batch effect if a feature can be predicted more accurately than random chance (AUROC of 0.5) with standard cross validation, but cannot be predicted with preserved site cross validation."

If you are saying a feature is batch correlated because it has some dependency on sample stratification, this is not necessarily due to batch effect.

See <https://www.sciencedirect.com/science/article/abs/pii/S1359644617304178>

Reviewer #2:

Remarks to the Author:

Howard et al have presented an interesting work of relevance for the field of deep learning H&E analysis. Strengths of the work are: 1) site-specific batch effects are a timely and important issue in deep learning-based H&E analysis for the better development of classifiers and survival models, 2) Relevant image features (RGB, Haralick) are thoroughly compared across centers, 3) The effects of several well-known color augmentation and normalization approaches are investigated for their impact on site-specific batch correction, and 4) An intelligent quadratic programming approach has been applied to correct for batch effects in TCGA image sets through optimization of sample grouping in the training of deep learning models.

The main result is that site-specific batch effects contribute non-trivially to image-based predictors of a variety of genotypes and phenotypes across 6 cancer tissue types and multiple sites. This result is somewhat to be expected, as there have not previously been image-analysis-oriented approaches for standardizing H&E staining procedures. So it is almost inevitable that there would be site-specific effects, though it was not obvious how much data augmentation and normalization would impact these results. The authors also explore how the clinical characteristics of the patient populations in each contributing site are entangled with the site specific staining variations.

Major Concerns

1) p.5: "one versus rest area under the receive operating characteristic (AUROC) curve". These are very high AUC values. It would be useful to know if the predictions are stemming from the biases in patient populations at each site or if they are from the slide staining procedure. The latter seems more likely, given that there is still some diversity in patient populations within each site. If so, one would expect that the color and Haralick distributions would have site-specific biases even in different histologies. Please address whether the site-specific behaviors shown in Fig 3 for breast have the same trends in other tissues.

2) My main critique of the work is that there is only minor discussion / interpretation of the biology of those features where site-specific effects make a big difference vs. those features where such effects are negligible. The authors have presented a little of this in Fig 6a and then buried most of the results in the supplemental tables. The Discussion considers a few aspects such as how young age is correlated with high grade tumors and how ethnicity is related to breast cancer histology. However, the authors are overly cautious about enumerating predictors of genetic features and whether we should believe them. More explanation along these lines would significantly strengthen the paper.

Notably, ref. 16 (Kather et al Nat Cancer 2020) is an important paper by some of the same investigative team. The authors should investigate the effect of site-specific batch effects on the strongly predicted genetic features in the relevant cancer types identified in ref 16. These should include the BRCA, COAD, HNSC, KIRC, LUAD, and LUSC genetic features meeting the p-value threshold in Figs 2, 3, and 4 of ref. 16.

p.6 and p.18: The methods for standard and preserved cross validation are not sufficiently described. For standard cross-validation, does this mean that data from all sites are merged and

then samples are chosen without respect to site such that the classes are balanced and 3 equal folds are produced? If so that should be stated directly. Currently on p.18 there is only a brief phrase: "standard cross validation - stratifying by site." The term "preserved site cross validation" is only alluded to with respect to a stratification procedure that needs to be stated explicitly (The only explanation is half a sentence: "site preserved cross validation – where each site is isolated to a single fold, and secondarily stratifying by site"). While this concern is straightforward, it is important that the stratification methods be described as clearly as possible in order for the results of the work to be effectively evaluated.

Minor:

p.5: "one versus rest area under the receive operating characteristic (AUROC) curve (Supplemental Table 2)" : should be Supp Table 3.

p.6: "Naturally, if a deep learning model can distinguish sites based on non-biologic differences between slide staining patterns and slide acquisition techniques, models will learn the clinical variability between samples at different sites. This is analogous to the Husky versus Wolf problem, where a deep learning model falsely distinguishes pictures of these two canines based on the fact that more wolves are pictured in snow":

These concepts are described inexactly. For example, in the text the phrase "models will learn the clinical variability" would more appropriately be replaced by "models meant to predict clinical variables will actually learn staining variability." Second, in the Husky versus Wolf problem, it is imprecise to say that the deep learning model "falsely distinguishes pictures of these two canines". Rather, the model learns predictive features that do not allow generalization to particular types of validation sets, but the distinguishing of pictures in the training set is not itself "false."

p.6: There is mixing of informal analysis results not supported by data together with method description ("It is not possible to perfectly stratify by race, but we can select partitions of sites as close as possible for k-fold cross-validation using convex optimization / quadratic programming").

p. 6: "demonstrated a clear basophilic-eosinophilic color gradient". This is not obvious as the tiles are very small. Please show labeled basophils / eosinophils in characteristic tiles in a magnified supplementary figure.

p.6: The choice of how to show p-values is distracting – it would be easier to evaluate the findings if the Benjamani-Hochberg-corrected p-values were shown instead of the uncorrected values.

Figure 6 caption: Background and subjective statements should be moved out of the caption to allow the reader to assess the findings objectively.

p.7: "an average decrease in AUROC of 0.064 (range, -0.012 to 0.309) was seen". I guessed that "average decrease" means the difference in AUCs for the standard and preserved site cross validation. But this should be actually stated.

p.12: "validation datasets are not frequently unavailable" -> "validation datasets are not frequently available"

p.13: "Indivumed"?

Reviewer #3:

Remarks to the Author:

Howad et al. discuss the batch effect on deep-learning model applied to digital histopathology,

using images from the TCGA database.

The number of deep-learning studies applied to histopathology images has significantly increased in the past few years. Most of them rely on images available from repositories such as TCGA.

Whole slide images from the TCGA come from different institutes. The authors show that in some cases, deep-learning method applied to such dataset may suffer from batch effect.

Clarity and context:

The article is well written, mostly clear and put into context.

Key results:

When doing deep-learning analysis, improper splitting of the dataset can lead to over-estimation of the performance due to batch effect. The impact is statistically significant for some of the features analyzed, but, as stated by the authors, the small decrease is not clinically relevant.

Originality / Significance:

The method is clearly explained and transparent.

The study is important in making the community more aware of such effect.

Through the analysis of batch effect, the paper raises interesting questions regarding the best way to balance datasets in deep-learning approaches. I appreciate the result section and analysis done.

Conclusions made:

In general, I feel the conclusions drawn could be more tempered.

- Lines 153-192:

The results are interesting but the comments on the supplemental table 5 are mostly done on the features and conditions that show a significant contamination by non-preserved site strategy.

There are also many cases where results are not significantly different, and in some cases when it depends on the slide adjustment method used, but this is only briefly mentioned. Some parts should also be more precise; for example: "The effect size is small for the majority of features, especially those with a clear histologic basis such as tumor histologic subtype and grade. The fact that performance decreased in nearly all models."

It is interesting but it would be more convincing, instead of using expressions like "the majority" and "nearly all", to use numbers to support the claims (proportion of predictable features that do show significant decrease in AUC, for example). Maybe a summary of table 5 might help: like for example, taking aside the features that can't be predicted by any approach, what is the proportion of features which AUC do significantly decrease or become random versus those not affected significantly?

- Line 202: "The fact that performance decreased in nearly all models (Figure 6a)"

Figure 6a: to which slide adjustment method does it correspond to? May be interesting to see such a panel with a color normalization method and one without, for sake of visual comparison.

- Line 208: "in some cases despite any form of normalization / augmentation." What would be your hypothesis/explaining why stain normalization and augmentation does not always help?

- Lines 226-230: not sure I understand this section. It seems to say that the following papers do not use external validations, but some of the citations seem to have at least some partial external validation. I think I misunderstand the point here? The paper seems to agree however that external cohort is crucial when per site splitting is not done.

- Line 289: why was resolution not integrated in the study? It seems that in the text, the authors often assume the per-site bias is due to the staining differences, but couldn't it be because of the resolution mis-match?

Suggested (optional) improvements:

- To follow up with the last comment, another factor that might participate in the identification of site may be the pixelsize. Pixel size of different scanners are slightly different. If you rescale the tiles such as the field of view and pixelsize are more similar to each other, would site-specific AUC still be observed?
- To differentiate whether predictions are associated with biological specificities or biased by staining/acquisition procedures, would a classifier solely based on clinical data (no images, just demography and description of the tumor characteristics) be able to predict the site of origin?
- I appreciate a lot that different color normalization strategies have been compared. To identify if one of the stain is driving more those differences, it may be interesting to also deconvolve the stains hematoxylin & eosin components and repeat part of the analysis on those.

Other minor comments/suggestion:

- Line 111: Sup table 2 is referred to - shouldn't it be sup table 3?
- Fig 5b: it is difficult to see much, it would be good to increase the resolution of the UMAP. Also, how does the UMAP looks like after the different color normalization strategies? Does it still show some kind of gradient? Is the gradient really stain related, or is it related to the density of cells and type of tissue?
- Line 264 "datasets are not frequently unavailable" did you mean available?
- Line 343: please detail how annotations were performed (which software / GUI, etc)
- (Optional) Figure 2-4 are done on breast. In supp figure, it might be interesting to show similar figures for just one other type of cancer.

Code:

- I greatly appreciate the code being made available. I would suggest to specify what version of python is used, and what hardware requirement are best. On a station equipped with GPU and CPU, the execution took me ~4 minutes instead of the 0.16 sec specified. Also, the result is slightly different from the one it is supposed to (I ran it 3 times and obtained the same result below – so I guess there's no "random" component that could explain the difference with the supposed result in the README file?):

Crossfold 1: A - 56 B - 246 Sites: ['Site 3', 'Site 5', 'Site 6', 'Site 9', 'Site 11', 'Site 12', 'Site 19', 'Site 27', 'Site 28', 'Site 30', 'Site 32']

Crossfold 2: A - 54 B - 244 Sites: ['Site 0', 'Site 1', 'Site 2', 'Site 4', 'Site 7', 'Site 10', 'Site 13', 'Site 20', 'Site 21', 'Site 25', 'Site 26', 'Site 36']

Crossfold 3: A - 52 B - 261 Sites: ['Site 8', 'Site 14', 'Site 15', 'Site 16', 'Site 17', 'Site 18', 'Site 22', 'Site 23', 'Site 24', 'Site 29', 'Site 31', 'Site 33', 'Site 34', 'Site 35', 'Site 37']

- Also, it is unclear whether this code was the one used for Sup Table 4. If so, can you please in the example specify the exact inputs required? It is unclear how you would operate to balance so many features with this code.

RESPONSE TO REFEREES

Dear Editors and Referees,

We would like to thank the editors and referees for their thorough review, and for providing very insightful comments. Our point by point response is provided below:

Reviewer #1 (Remarks to the Author):

The Impact of Digital Histopathology Batch Effect on Deep Learning Model Accuracy and Bias

Overall comments:

I like the premise. In fact, I had a discussion with the editor of Nature MI on the importance of batch effects on ML performance. The authors touch on a very important topic, one that has clear implications on how AI/ML can be adapted for use on clinical data.

However, I feel that the authors do not have a good sense of what batch effects are --- they seem to confuse clinical and subpopulation effects as batch effects (which are technical biases).

They also attribute the performance of a feature as “batch correlated” if a feature provides good performance in random data as opposed to the preserved site cross validation --- this is not necessarily due to the influence of a batch effect.

I think there is value in carefully and repeatedly stratifying sample to carefully test for the confounding effects of some variable. But the weakness of such an approach is that the confounding variable must be known and described in the meta-data in order for it to be evaluated and tested. It is also messy and tedious to perform.

I also think there is limited value in saying generally about idiosyncratic batch effect, which is confounded with other biological/clinical subfactors, and then saying that stratification can help minimize such effects. I think it is more useful if a more controlled experiment can be used to illustrate the impact of the batch effect on ML given some assumptions about its nature and distributions.

We appreciate your astute and insightful commentary about the potential etiologies of what we had described as batch effect in TCGA. There are clearly clinical and subpopulation variability between the sites of interest in TCGA which may be detectable inherently from histology, and these components of batch effect are inextricably intertwined – and as you described the ability to carefully tease these apart is not straightforward and requires knowledge about meta-data that is unavailable in TCGA.

However, we believe there is indirect evidence to suggest that for many of the features evaluated, the ‘batch correlation’ we identified is predominantly due to batch effect rather than subpopulation differences, although we have updated our paper with careful wording to make the reader aware of this potential confounding factor which may be contributing as well.

First, the tissue submitting sites are nearly perfectly predicted by our deep learning model – moreso than any standard histologic feature – across all disease types queried. As 1) no other biologic feature was detected to the same degree of accuracy directly from histology and 2) from our experience and review of the literature there is no report that disease histology / biology varies so dramatically from region to region that the submitting region could be identified solely from the true biologic histology alone. We also created a similar neural network prediction model using all the clinical / demographic / genomic features studied to predict site, achieving average AUROCs in the 0.62 range, nowhere near the > 0.90 range seen in our histology models.

Furthermore, models trained to identify features of interest weight slide staining features heavily in their decision making. For example, the UMAP seen in Figure 5b clearly illustrates that staining differences is a

principle component to the determination of tissue submitting site, given the clear hematoxylin / eosin gradient seen. A similar pattern is seen with UMAPs for other 'batch correlated' features such as ethnicity (Supplemental Figure 4b) but individual sites are less widely separable for features that are less correlated with site such as BRCA (supplemental figure 4c). Additionally, the degree of decline in AUROCs with preserved site validation, in some cases on the order of 0.20 or higher, would suggest that there is something other than biologic covariates at play. Since the most highly detected biologic features reach an AUROC of 0.7 – 0.8, a drop in magnitude of AUROC by 0.20 solely due to clinical / subpopulation differences would require that a feature of interest be perfectly correlated with a true biologic covariate, which is unlikely to be true.

We have attempted to produce a more highly controlled experiment as per your recommendation to more precisely study the impact of a highly detectable histologic covariate, versus stain related batch effect, on detection of a hypothetical feature (Supplemental Figure 5, Supplemental Table 4). The magnitude of change in AUROC associated with a true biologic covariate versus a batch effect would support that at least some of our findings are due to true batch effect. Additionally, we demonstrate that at moderate – high degrees of stain related differences between slides, the impact of biologic covariates on performance is minimized.

Ultimately, when attempting to address 'batch effect' for predictive models in digital histology, the goal is to identify features that are solely present due to idiosyncracies of a dataset, and those that are likely to have an underlying histologic basis that is likely to be replicated in other datasets. We believe site preserved cross validation may be a useful screening tool – whether the decline in performance is due to true batch effect or subpopulation clinical differences that are not reflective of a larger population. However, we certainly agree we have been less specific in our terminology, and have rectified this throughout the manuscript, added significant discussion of the potential of clinical covariates. We appreciate your continued review and any further advice you have on the topic.

I have the following specific comments:

“Furthermore, ethnicity can also be inferred from histology due to site-level batch effect, which must be accounted for to ensure equitable application of DL.”

May not be batch effect. Unless the batch factor is also confounded with ethnicity information.

Thank you, we have removed the attribution of this finding to pure batch effect.

“Naturally, if a deep learning model can distinguish sites based on non-biologic differences between slide staining patterns and slide acquisition techniques, models will learn the clinical variability between samples at different sites.”

This happens regardless of whether the non-biologic difference exists or not.

Thank you – we have specified that this can also be due to subpopulation differences (after our discussion of the Husky vs Wolf problem – to allow for better flow).

“First, stratifying by feature of interest and ignoring site, a number of features may be predicted due to batch effect between sites. We can correct for biased results by ensuring sites are isolated to single data fold, or preserved site cross validation.”

This expression is convoluted. What I think it sounds like to me is that you are comparing two groups where no biologic differences exist except the batch factor. In other words, you are looking for batch-correlated variables.

But what happens if a feature is both class and batch confounded? Do you reject or drop it anyway?

We have clarified this expression to highlight that some features may only be detectable in a multi-institutional dataset due to batch effect or subpopulation differences would be abrogated with this method. We think this is an imperfect but clinically relevant approach, as subpopulation differences can also lead to a lack of external validity if not reflected in the larger cancer population.

“However, it is still necessary to stratify by the variable of interest, as stratification improves the variance and bias of estimates of accuracy³⁸”

This follow up sentence is weird. You already said to stratify by feature of interest. And now you say it is still necessary to do this?

I also think direct-copy statements like “stratification improves variance and bias of estimates of accuracy” is not helpful. You mean to say it reduces variance and bias.

Thank you for the comment – we have reworded these sentences. What we’re trying to convey, is that if you naively take 30 sites and split them into 3 piles of 10, then it is likely that your feature of interest will be imbalanced across the 3 piles. If we randomly select sites for preserved-site cross validation, this imperfect stratification would potentially explain the decreased AUROCs seen with this method of cross validation. So perfect (or close to perfect) stratification would suggest that the decline in AUROC with preserved site validation is not solely due to poor stratification.

“It is not possible to perfectly stratify by race, but we can select partitions of sites as close as possible for k-fold cross-validation using convex optimization / quadratic programming³⁹”

I think having multiple partitions based on single variable splits is not very useful.

Thank you for the comment – we are unsure what is meant by multiple partitions on single variable splits – but have reworded this phrasing.

“Essentially, we optimize an error represented by the deviation of a validation fold from perfect stratification, where the percentage of patients in each subgroup is equal to the average percent over the dataset.”

This is a very convoluted way of simply saying --- the proportion of patients in each fold is kept constant.

So you are trying to adjust for situations where your fold is not representative of the label proportions of the original data.

But you also said that for certain features, it is not possible to get a good stratification.

Thank you for pointing this out - we have clarified this language. We were attempting to describe the method in which convex optimization can be used to solve this problem – i.e. the error function that would be required – but this is probably better defined mathematically in the methods as we have done, with clearer language in the results section as you have pointed out. Although our method of cross validation does not always yield good results – we have noted this as a limitation, and in the majority of cases the stratification is identical to that which would be obtained with standard cross validation.

And also, how do you know that all meaningful and important features that biases your model are recorded?

As you pointed out at first, I think it is not possible to know that every feature that impacts model performance is accounted for, as only a fraction of relevant features are available for patients in TCGA. The purpose of our

method of cross validation is to maximize the likelihood of external validity; by splitting patients into folds irrespective of site, it runs the risk of both batch effect and subpopulation related biases specific to a site leading to an artificially elevated model performance. Since it is not always possible to know the subpopulation differences present in a single site – by training models on one site and testing performance on another site, anything learned from either batch effect or subpopulation differences that is not represented in the larger population will no longer be able to influence model performance.

Also, no mention of the model type you use? Maybe the outcome is an artifact of the model or hyperparameter settings?

Thank you – we have added a reference to our model type in the body of the text, but detailed description of the model and hyperparameters are listed in the methods.

“We applied this method of stratification to all outcomes listed in Figure 2 and Supplemental Table 1.”

I am confused here. First you say you are optimizing a method that accounts for deviation from sample proportions.

Now you are saying that your method is a stratification approach.

We apologize for the confusing wording – we were using stratification in the non-statistical sense – i.e. a method of dividing objects into groups; but we can see this would create confusion with statistical stratification and have changed our wording.

“For 49 features evaluated, an average decrease in AUROC of 0.064 (range, -0.012 to 0.309) was seen, with a number of models demonstrating a meaningful decrease in performance attributable to batch effect related bias (Figure 6a, Supplemental Table 5).”

I think whenever a drop in performance is observed, it may not necessarily be attributed to just “batch”.

Although we feel there is evidence to suggest that batch effect is a significant contributor – we have altered the language to reflect this clarification.

“...for TCGA-LUSC declined significantly with preserved site cross validation regardless of stain normalization / augmentation.”

So this means stain norm and augment did not work. But made things worse.

But I am also confused, you said you assessed which models had the significant decline. But here, you only allude to one particular example.

We have provided summary statistics to describe how the decline in performance with preserved site validation is affected by forms of stain normalization. Of note, although ~33% of features become unpredictable without normalization and preserved site validation, only about ~20% of features that were predictable at baseline are unpredictable once a normalization technique is applied. Thus, normalization may reduce the dependence of models on site specific characteristics in making predictions, and allow it to weight histologic characteristics more heavily. However, there are still features that decline significantly when training on one group of sites and testing on another – suggesting a lack of generalizability.

“..identification of a feature of interest from histology is ostensibly solely due to batch effect if a feature can be predicted more accurately than random chance (AUROC of 0.5) with standard cross validation, but cannot be predicted with preserved site cross validation.”

If you are saying a feature is batch correlated because it has some dependency on sample stratification, this is not necessarily due to batch effect.

See <https://www.sciencedirect.com/science/article/abs/pii/S1359644617304178>

We have removed this wording as per your comments. We agree that dependency on stratification is not solely due to batch effect – however, if there are unknown features that we have not accounted for in stratification, we would expect that using preserved site cross validation would in some cases improve performance – instead of consistently decreasing model performance as we have shown.

Reviewer #2 (Remarks to the Author):

Howard et al have presented an interesting work of relevance for the field of deep learning H&E analysis. Strengths of the work are: 1) site-specific batch effects are a timely and important issue in deep learning-based H&E analysis for the better development of classifiers and survival models, 2) Relevant image features (RGB, Haralick) are thoroughly compared across centers, 3) The effects of several well-known color augmentation and normalization approaches are investigated for their impact on site-specific batch correction, and 4) An intelligent quadratic programming approach has been applied to correct for batch effects in TCGA image sets through optimization of sample grouping in the training of deep learning models.

The main result is that site-specific batch effects contribute non-trivially to image-based predictors of a variety of genotypes and phenotypes across 6 cancer tissue types and multiple sites. This result is somewhat to be expected, as there have not previously been image-analysis-oriented approaches for standardizing H&E staining procedures. So it is almost inevitable that there would be site-specific effects, though it was not obvious how much data augmentation and normalization would impact these results. The authors also explore how the clinical characteristics of the patient populations in each contributing site are entangled with the site specific staining variations.

Major Concerns

1) p.5: “one versus rest area under the receive operating characteristic (AUROC) curve”. These are very high AUC values. It would be useful to know if the predictions are stemming from the biases in patient populations at each site or if they are from the slide staining procedure. The latter seems more likely, given that there is still some diversity in patient populations within each site. If so, one would expect that the color and Haralick distributions would have site-specific biases even in different histologies. Please address whether the site-specific behaviors shown in Fig 3 for breast have the same trends in other tissues.

Thank you for raising this comment – we agree that the latter is most likely – and have described the variance in color and Haralick distributions, and replicated Fig 3 for other histologies as supplemental figure 2. Additionally, the variance in these distributions with different forms of normalization are described for other histologies in supplemental figure 3.

2) My main critique of the work is that there is only minor discussion / interpretation of the biology of those features where site-specific effects make a big difference vs. those features where such effects are negligible. The authors have presented a little of this in Fig 6a and then buried most of the results in the supplemental tables. The Discussion considers a few aspects such as how young age is correlated with high grade tumors and how ethnicity is related to breast cancer histology. However, the authors are overly cautious about enumerating predictors of genetic features and whether we should believe them. More explanation along these lines would significantly strengthen the paper.

Notably, ref. 16 (Kather et al Nat Cancer 2020) is an important paper by some of the same investigative team. The authors should investigate the effect of site-specific batch effects on the strongly predicted genetic features in the relevant cancer types identified in ref 16. These should include the BRCA, COAD, HNSC, KIRC, LUAD, and LUSC genetic features meeting the p-value threshold in Figs 2, 3, and 4 of ref. 16.

We appreciate the comments. We have provided a more structured approach to the discussion of why some features are greatly impacted by site specific effects and others are not. Demographic features are highly impacted, whereas known histologic features rarely experience a decline. Some of these non-intuitive features that may have a histologic basis such as genetics appear to lie in the middle, with a proportion having an impact of batch effect.

We agree that further analysis of significant features in this prior work from Kather et al deserve further exploration. For symmetry's sake, we have been performing analysis for all features with both standard and site preserved cross validation both at baseline and with six different forms of stain normalization / augmentation to demonstrate the consistency of batch effect for analyzed features, effectively increasing the computational difficulty for feature analysis by a magnitude of 14 from the referenced paper. As such, we have selected a subset of 10 highly predicted mutations in this prior work and analyzed them as requested. We think this will strike a balance between providing readers with additional trust in certain genetic findings (and raising question to others that were not as well predicted in our analysis) – while still demonstrating the consistency with forms of stain normalization. Further analysis of the residual genes with preserved site validation is in process (without varying the methods of stain normalization / augmentation) – but we feel this is sufficiently different from the hypothesis of this paper that it would be best included in separate work. Additionally, as the number of requested features have continued to grow after initial conception – we have modified our statistical analysis to include bootstrapping as per the referenced Kather et al paper. We have disclosed this modification as a limitation in our discussion.

p.6 and p.18: The methods for standard and preserved cross validation are not sufficiently described. For standard cross-validation, does this mean that data from all sites are merged and then samples are chosen without respect to site such that the classes are balanced and 3 equal folds are produced? If so that should be stated directly. Currently on p.18 there is only a brief phrase: “standard cross validation - stratifying by site.” The term “preserved site cross validation” is only alluded to with respect to a stratification procedure that needs to be stated explicitly (The only explanation is half a sentence: “site preserved cross validation – where each site is isolated to a single fold, and secondarily stratifying by site”). While this concern is straightforward, it is important that the stratification methods be described as clearly as possible in order for the results of the work to be effectively evaluated.

Thank you for your comment – you have correctly surmised how we did standard cross validation – and we have explicitly stated this in the methods section as you have worded. Additionally, we have explicitly stated that our preserved site stratification procedure is generated as a solution to the convex optimization problem listed, and with the software provided.

Minor:

p.5: “one versus rest area under the receive operating characteristic (AUROC) curve (Supplemental Table 2)” : should be Supp Table 3.

Thank you, we have corrected this.

p.6: “Naturally, if a deep learning model can distinguish sites based on non-biologic differences between slide staining patterns and slide acquisition techniques, models will learn the clinical

variability between samples at different sites. This is analogous to the Husky versus Wolf problem, where a deep learning model falsely distinguishes pictures of these two canines based on the fact that more wolves are pictured in snow”:

These concepts are described inexactly. For example, in the text the phrase “models will learn the clinical variability” would more appropriately be replaced by “models meant to predict clinical variables will actually learn staining variability.” Second, in the Husky versus Wolf problem, it is imprecise to say that the deep learning model “falsely distinguishes pictures of these two canines”. Rather, the model learns predictive features that do not allow generalization to particular types of validation sets, but the distinguishing of pictures in the training set is not itself “false.”

Thank you for these clarifications – you are correct that we are inexact in our wording for both and we have provided additional comments in the text to address.

p.6: There is mixing of informal analysis results not supported by data together with method description (“It is not possible to perfectly stratify by race, but we can select partitions of sites as close as possible for k-fold cross-validation using convex optimization / quadratic programming”).

Thank you - We have removed this informal description – and have stuck to the formal discussion of performance metrics of preserved site cross validation.

p. 6: “demonstrated a clear basophilic-eosinophilic color gradient”. This is not obvious as the tiles are very small. Please show labeled basophils / eosinophils in characteristic tiles in a magnified supplementary figure.

Thank you for the comment – we were referring to hematoxylin and eosin rich staining patterns, rather than the infiltrate of specific cells. We have highlighted this in the figure and removed the use of the word basophilic to avoid confusion.

p.6: The choice of how to show p-values is distracting – it would be easier to evaluate the findings if the Benjamini-Hochberg-corrected p-values were shown instead of the uncorrected values.

We have updated our text to use FDR corrected p-values in all figures.

Figure 6 caption: Background and subjective statements should be moved out of the caption to allow the reader to assess the findings objectively.

Thank you, we have attempted to remove subjective language from this caption.

p.7: “an average decrease in AUROC of 0.064 (range, -0.012 to 0.309) was seen”. I guessed that “average decrease” means the difference in AUCs for the standard and preserved site cross validation. But this should be actually stated.

Thank you – I have added this description.

p.12: “validation datasets are not frequently unavailable” -> “validation datasets are not frequently available”

Thank you – this is corrected.

p.13: “Indivumed”?

This is a company that contributed a significant number of slides to the TCGA-COAD/READ cohorts. We have provided additional description to clarify in the text.

Reviewer #3 (Remarks to the Author):

Howad et al. discuss the batch effect on deep-learning model applied to digital histopathology, using images from the TCGA database.

The number of deep-learning studies applied to histopathology images has significantly increased in the past few years. Most of them rely on images available from repositories such as TCGA. Whole slide images from the TCGA come from different institutes. The authors show that in some cases, deep-learning method applied to such dataset may suffer from batch effect.

Clarity and context:

The article is well written, mostly clear and put into context.

Key results:

When doing deep-learning analysis, improper splitting of the dataset can lead to over-estimation of the performance due to batch effect. The impact is statistically significant for some of the features analyzed, but, as stated by the authors, the small decreased is not clinically relevant.

Originality / Significance:

The method is clearly explained and transparent.

The study is important in making the community more aware of such effect.

Through the analysis of batch effect, the paper raises interesting questions regarding the best way to balance datasets in deep-learning approaches. I appreciate the result section and analysis done.

Conclusions made:

In general, I feel the conclusions drawn could be more tempered.

• **Lines 153-192:**

The results are interesting but the comments on the supplemental table 5 are mostly done on the features and conditions that show a significant contamination by non-preserved site strategy. There are also many cases where results are not significantly different, and in some cases when it depends on the slide adjustment method used, but this is only briefly mentioned. Some parts should also be more precise; for example: “The effect size is small for the majority of features, especially those with a clear histologic basis such as tumor histologic subtype and grade. The fact that performance decreased in nearly all models.”

It is interesting but it would be more convincing, instead of using expressions like “the majority” and “nearly all”, to use numbers to support the claims (proportion of predictable features that do show significant decrease in AUC, for example). Maybe a summary of table 5 might help: like for example, taking aside the features that can't be predicted by any approach, what is the proportion of features which AUC do significantly decrease or become random versus those not affected significantly?

Thank you – we have provided a summary table to describe the changes in Supplemental Table 6 (Supplemental Table 7), and have provided a more concrete description of performance in the text.

• **Line 202: “The fact that performance decreased in nearly all models (Figure 6a)”**

Figure 6a: to which slide adjustment method does it correspond to? May be interesting to see such a panel with a color normalization method and one without, for sake of visual comparison.

Thank you – Fig 6a represents a panel without color normalization. I have added an additional panel with grayscale normalization (in which the number of features with decrease was the smallest) for visual contrast (Fig 6b).

• **Line 208: “in some cases 207 despite any form of normalization / augmentation.” What would be your hypothesis/explaining why stain normalization and augmentation does not always help?**

Thanks - we have provided some further discussion of this. Without stain normalization, ~33% of features become undetectable, whereas with forms of stain normalization, only about ~20% of detectable features become undetectable. Thus, we believe when the first / second order image characteristics that differ between sites become less pronounced, they may be less easily detectable by models, as opposed to true histologic differences. However, as per Figure 3/4 and supplemental figures 2/3 first / second order image characteristics remain distinguishable between sites, and so we suspect models can still detect some stain related abnormalities. As requested by reviewer 1, we did an additional experiment exploring the impact of subpopulation differences vs staining on predictions – and we found that although predictive accuracy declined slightly with stain normalization / augmentation (similar to our studies of site prediction), an artificial stain variation remained highly detectable despite normalization / augmentation.

• **Lines 226-230: not sure I understand this section. It seems to say that the following papers do not use external validations, but some of the citations seem to have at least some partial external validation. I think I misunderstand the point here? The paper seems to agree however that external cohort is crucial when per site splitting is not done.**

We do attempt to identify papers that describe some subset of findings that have not been externally validated – but this was not clear in our text and we have hopefully address that. A number of our citations such as Kather et al, Fu et al, Scmauch et al externally validate some but not all findings – indeed these papers all describe models developed on large datasets with over 10 cancer subtypes – and it would be impractical / impossible to validate each finding listed. Our intent is merely to suggest that preserved site cross validation could be an additional test of validity for when external datasets are not available, as certainly some of these features will not stand the test of external validation. For example, the Fu et al paper – which externally validated only their breast cancer findings – noted that up to 32% of genetic features declined to an AUROC of < 0.5 with external validation – a similar proportion of findings that became insignificant with preserved site cross validation in our study.

• **Line 289: why was resolution not integrated in the study? It seems that in the text, the authors often assume the per-site bias is due to the staining differences, but couldn't it be because of the resolution mis-match?**

Thank you - we agree that there may be other factors contributing to the per site bias – and have incorporated them into Figure 1 and our discussion. Indeed – this may be one of the reasons for the above comment about line 208 – why normalization / augmentation do not resolve all the elements of batch effect. All tiles were created from source slides at an equal pixel per micrometer resolution (299px:302um) so this should minimize resolution related biases; but it is true that a compression artifact may be introduced when converting tiles to a specified resolution. We believe preserved site cross validation will help mitigate biases that result from resolution associated differences, but separating them from stain related differences between sites is a challenge.

Suggested (optional) improvements:

• **To follow up with the last comment, another factor that might participate in the identification of site may be the pixelsize. Pixel size of different scanners are slightly different. If you rescale the tiles such as the field of view and pixelsize are more similar to each other, would site-specific AUC still be observed?**

As above, we believe we have addressed the resolution issue partially by extracting tiles at a fixed pixel per micrometer size. However, there are certainly artifacts related to differences in resolution / compression that are unaccounted for – but it may be difficult to tease these features apart.

• **To differentiate whether predictions are associated with biological specificities or biased by**

staining/acquisition procedures, would a classifier solely based on clinical data (no images, just demography and description of the tumor characteristics) be able to predict the site of origin?

We have created a classifier using a multilayer perception model with one 500 node hidden layer (equivalent to the final layer we have added to the end of our Xception model), and provided predictive accuracy for such a model in comparison to our results. Ultimately the performance was much poorer than was seen from the results with our histologic model, likely because batch related features are contributing.

• I appreciate a lot that different color normalization strategies have been compared. To identify if one of the stain is driving more those differences, it may be interesting to also deconvolve the stains hematoxylin & eosin components and repeat part of the analysis on those.

We agree this would be another interesting comparison to perform but if reasonable we may defer this for a future project as we are unsure we can give such an analysis justice in the context of this study, as it would likely require a significant number of models trained to draw firm conclusions about the relative importance of the two stain components.

Other minor comments/suggestion:

• Line 111: Sup table 2 is referred to - shouldn't it be sup table 3?

Thank you for catching this, we have corrected it!

• Fig 5b: it is difficult to see much, it would be good to increase the resolution of the UMAP. Also, how does the UMAP looks like after the different color normalization strategies? Does it still show some kind of gradient? Is the gradient really stain related, or is it related to the density of cells and type of tissue?

Thank you – we have provided a higher resolution version of this image, and the predominant effect of the gradient visualized appears to be stain related rather than cellularity from our interpretation. If an even high resolution image would be helpful we can provide such an image. We also provided UMAP with stain normalization, and for other example outcomes of interest in Supplemental Figure 4.

• Line 264 “datasets are not frequently unavailable” did you mean available?

Thank you, we have fixed this typo.

• Line 343: please detail how annotations were performed (which software / GUI, etc)

Thank you – we have provided this clarification (annotations were performed in QuPath v0.12).

• (Optional) Figure 2-4 are done on breast. In supp figure, it might be interesting to show similar figures for just one other type of cancer.

Thank you, we have provided in supplemental figures some representative figures for other cancer subtypes.

Code:

• I greatly appreciate the code being made available. I would suggest to specify what version of python is used, and what hardware requirement are best. On a station equipped with GPU and CPU, the execution took me ~4 minutes instead of the 0.16 sec specified. Also, the result is slightly different from the one it is supposed to (I ran it 3 times and obtained the same result below – so I guess there's no “random” component that could explain the difference with the supposed result in the README file?):

Crossfold 1: A - 56 B - 246 Sites: ['Site 3', 'Site 5', 'Site 6', 'Site 9', 'Site 11', 'Site 12', 'Site 19', 'Site 27', 'Site 28', 'Site 30', 'Site 32']

Crossfold 2: A - 54 B - 244 Sites: ['Site 0', 'Site 1', 'Site 2', 'Site 4', 'Site 7', 'Site 10', 'Site 13', 'Site 20',

'Site 21', 'Site 25', 'Site 26', 'Site 36']

Crossfold 3: A - 52 B - 261 Sites: ['Site 8', 'Site 14', 'Site 15', 'Site 16', 'Site 17', 'Site 18', 'Site 22', 'Site 23', 'Site 24', 'Site 29', 'Site 31', 'Site 33', 'Site 34', 'Site 35', 'Site 37']

Thank you for pointing this out. We have specified the version and hardware used for testing. We believe the difference in performance and results is related to the non-deterministic nature of CPLEX searches with multiple possible solutions. In particular, hardware chip instruction set, operating system, and compiler can alter CPLEX results even with the same 'random' initial state used for a solve. CPLEX provides details on this limitation here: <https://www.ibm.com/support/pages/note-reproducibility-cplex-runs>. Upon our investigation, this is likely why running this code on different platforms have different results. We have added a disclaimer to our Github page regarding the potential for different results and performance when run on different platforms. CPLEX lists a few parameters to minimize this variance – including feasibility / optimality tolerances and Markowitz tolerance – but even when setting these parameters to their optimal settings we still get slightly different results when running on different hardware, so unfortunately there may be some variance hindering reproducibility of CPLEX results.

Additionally, CPLEX will stop its search when it arrives at an optimal solution, and continue otherwise to a pre-specified time limit, which is an input parameter to our “generate” function. In some hardware settings / search states, our example dataset will not readily converge to an optimal solution, leading to a larger computational time, which is likely why we have different results in computational time.

• Also, it is unclear whether this code was the one used for Sup Table 4. If so, can you please in the example specify the exact inputs required? It is unclear how you would operate to balance so many features with this code.

Thank you – we did use the code to generate supplemental table 4. We have detailed how this was done in our Github. In short – we did not balance over all features simultaneously; these were the results of PreservedSite validation in attempting to balance each individual feature of interest across multiple categories. It would be relatively straightforward from our code to create a similar tool to stratify across multiple outcomes of interest with preserved site cross validation, although performance would likely decline – we would be happy to draft this if of interest.

Reviewers' Comments:

Reviewer #1:

Remarks to the Author:

the authors have responded satisfactorily to the points raised.

Reviewer #2:

Remarks to the Author:

The authors have thoroughly addressed my concerns. Notably, the new supplemental Figure 2 is informative. For those cases where pairs of contributing centers can be compared across two tissue types, there do seem to be trends in the statistical features observed between centers (e.g. GPCC vs Indivumed between a and b; to a lesser extent Pittsburgh vs MDACC in c and d; IGC vs Asterland in e and f). There are some tissue-specific behaviors, but some deviation is not unexpected as cell type differences across tissues are likely to be a confounding factor. Also, the new text in the "Impact of Site-Specific Digital Histology Signatures on Deep Learning Model Performance" is a significant improvement to the manuscript and addresses my concerns about the prediction of genetic features. I would recommend acceptance of the manuscript, which will be valuable for many researchers in the field of deep learning-based histological image analysis.

-Jeff Chuang

Reviewer #3:

Remarks to the Author:

I think the authors have properly addressed the comments and made the manuscript stronger.

I just have one minor question regarding the new data on the mutations (Lines 234-241 - Line 274-277). It looks like mutations which are no longer predictable tend to also be mutations which are slightly more rare and where the datasets are more imbalanced.

If the authors think this is a factor that matters and that with potentially more data, the per site sensitivity would be lowered, then a small comment on it on the manuscript may be welcome.

RESPONSE TO REFEREES

Dear Editors and Referees,

We would like to thank the editors and referees for their thorough review, and for providing very insightful comments. Our point by point response is provided below:

Editorial Comments:

Please amend each instance of the concept of "race" and "racial" with the correct wording. If you are referring to genetics-inferred subpopulations, please use the term "ancestry" as you do in other cases. If you are referring to self-identified groups, then please use the term "ethnicity" ("ethnic"). See eg lines 251-263, in which all 3 terms are used, seemingly as if they were inter-exchangeable.

Thank you for identifying this inconsistency. We reviewed our manuscript and ensured that these terms are appropriately applied in a consistent fashion throughout.

We congratulate you on providing a sensible overview on the issues related to ethnicity and biological and non-biological factors causing differences in presentation and prognosis (lines 278-294). We would suggest maybe adding a sentence asking for more research to disentangle the two (biological and non-biological factors). We would suggest changing the example you mention in lines 291-294, in which it sounds like using data from a disadvantaged population (it is unclear also what "African patients" are in this example) might affect negatively the "privileged" one. We understand the argument but we would argue, at the same time, that it would be more sensitive to offer the opposite example.

Thank you – we completely agree it is better to highlight how deep learning can amplify disparities in disadvantaged populations. In lines 291-294 we were attempting to refer to the fact that if a site solely enrolls a disadvantaged population that experiences disease recurrence at a higher rate, then even the good prognosis patients from that disadvantaged population would be mislabeled if a model bases decisions on the overall characteristics of the site in question, leading to unnecessarily aggressive treatment. We also agree that more research is needed to clarify the individual contributions of biologic and non-biologic factors to racial disparities. We have reworded as follows:

Contributing factors may include delays in treatment initiation and inadequate intensity of therapy, however more research is needed to disentangle the biologic and non-biologic factors contributing to racial disparities in prognosis. Furthermore, as deep learning models are able to infer ancestry (and therefore race) from histology, they must be carefully implemented in an equitable fashion to avoid amplifying pre-existing inequities in cancer care. For example, if a prognostic model has learned the staining pattern of a site that recruits disadvantaged patients with a higher than average rate of recurrence, it may falsely denote all patients at that site as 'poor prognosis', which could lead to inappropriate administration of chemotherapy or other treatments to even the low risk patients in a disadvantaged population.

Reviewer #1 (Remarks to the Author):

the authors have responded satisfactorily to the points raised.

Reviewer #2 (Remarks to the Author):

The authors have thoroughly addressed my concerns. Notably, the new supplemental Figure 2 is informative. For those cases where pairs of contributing centers can be compared across two tissue types, there do seem to be trends in the statistical features observed between centers (e.g. GPCC vs

Indivumed between a and b; to a lesser extent Pittsburgh vs MDACC in c and d; IGC vs Asterland in e and f). There are some tissue-specific behaviors, but some deviation is not unexpected as cell type differences across tissues are likely to be a confounding factor. Also, the new text in the “Impact of Site-Specific Digital Histology Signatures on Deep Learning Model Performance” is a significant improvement to the manuscript and addresses my concerns about the prediction of genetic features. I would recommend acceptance of the manuscript, which will be valuable for many researchers in the field of deep learning-based histological image analysis.

-Jeff Chuang

Reviewer #3 (Remarks to the Author):

I think the authors have properly addressed the comments and made the manuscript stronger.

I just have one minor question regarding the new data on the mutations (Lines 234-241 - Line 274-277). It looks like mutations which are no longer predictable tend to also be mutations which are slightly more rare and where the datasets are more imbalanced.

If the authors think this is a factor that matters and that with potentially more data, the per site sensitivity would be lowered, then a small comment on it on the manuscript may be welcome.

Thank you for bringing up this point – we do agree that the mutation prediction is a challenging topic due to the rarity which in and of itself could create instability in predictive accuracy (rather than being a batch effect mediated phenomenon), and we have listed this in our discussion of mutations.

It must be noted that the prevalence of some of the genomic alterations evaluated in this study were rare, and thus they may be more susceptible to changes in predictive accuracy just due to random chance rather than from site specific digital histology signatures.

We agree that predictive accuracy for RNF43 may be affected by imbalance, and we do call attention to this as below, although the other poorly predicted genetic features were well stratified.

Our method of generating preserved site cross folds successfully stratified patients by outcomes of interest for the majority of features examined, but there were some notable outliers in the TCGA-COADREAD dataset.

Reviewers' Comments:

Reviewer #4:

Remarks to the Author:

The manuscript by Howard provides interesting observations on the impact of site-specific digital histology signatures on deep learning modeling accuracy. The focus of this critique centers on the author's treatment or approach to understanding the contribution of demographics to the site-specific digital histological signature. Central to this discussion is the author's statement:

"We have demonstrated that deep learning models have the potential to predict genomic ancestry with a high degree of accuracy from histology directly from learned patterns about the demographic make-up of the submitting sites rather than from intrinsic biology".

This statement and its implication require the most scrutiny. My concern is that our staging and grading system are established by the pathologist's semi-quantitative interpretations that are ultimately converted into "bins" or "cut-offs". Where the cut-offs are determined (e.g. grade and tumor size) based on the review of clinical outcome data assembled from years of data collected with little concern or statistical weight given to diversity/demographics. The traditional assumption is that "one size (cutoff) fits all". Thus, it is not unreasonable that there will be a recognizable heterogeneity within the bins based on RACE/Demographics that would only be measurable/discernible by the granularity of deep learning. This is then complicated "downstream" by the confounding linkages of tumor stage/grade to outcome-based on access to care (ie demographics).

Thus, there will be differences in histology and stage (grade+size) that will vary by time of presentation (note that tumor size will influence sampling and the degree of tumor heterogeneity recovered at the pre-analytic stage). Here in the remaining paragraph, is where I feel is the greatest impact based on demography and are major contributors to both disease outcome and what also is interpreted as tumor biology (particularly concerning size). Contributions will include known ancestry-linked differences in the frequency of TNBC (2-fold higher in blacks) which occurs at a younger age and has higher grade. Thus, all of these features of demographics can't be disentangled from their impact on histology, particularly given the established ability of deep-learning to discern hormone receptor status. Thus, I am sure that the deep-learning signature is picking up features of age association with grade; race association with receptor status; higher grade associated with TNBC; lower age associated with TNBC; and possible differences in tumor sampling (heterogeneity) that could also be linked to differences in tumor size at presentation. All of these features are invariably linked to demographics. The quantitative (discretized) nature of this linkage to intrinsic tumor biology cannot be recognized by the pathologists without the assistance of AI, which provides the granularity that enables the quantitative distinction.

Thus the authors should take a more nuanced approach to this statement and those later derived from it in the writing. While the signature can make accurate ancestry predictions, it is not clear these predictions are un-coupled from intrinsic biology.

One other concern particularly given the high percentage of patient of African ancestry submitted from one site (Chicago) is that the authors in their figures show the total number of patients contributed by each site so it is clearer what the full numeric impact of the demographic factors from this site has to the signature. Finally, it might be very interesting to have the authors present confusion matrices comparing ancestry predictions from whole slide images vs genomic ancestry calls for the overall TCGA population and a few example sites. This would give readers a much more concrete idea of accuracy with which models can predict genomic ancestry

RESPONSE TO REFEREES

Dear Editors and Referees,

We would like to thank the editors and referees for their thorough review, and for providing very insightful comments. Our point by point response is provided below:

The manuscript by Howard provides interesting observations on the impact of site-specific digital histology signatures on deep learning modeling accuracy. The focus of this critique centers on the author's treatment or approach to understanding the contribution of demographics to the site-specific digital histological signature. Central to this discussion is the author's statement:

"We have demonstrated that deep learning models have the potential to predict genomic ancestry with a high degree of accuracy from histology directly from learned patterns about the demographic make-up of the submitting sites rather than from intrinsic

biology”.

This statement and its implication require the most scrutiny. My concern is that our staging and grading system are established by the pathologist’s semi-quantitative interpretations that are ultimately converted into “bins” or “cut-offs”. Where the cut-offs are determined (e.g. grade and tumor size) based on the review of clinical outcome data assembled from years of data collected with little concern or statistical weight given to diversity/demographics. The traditional assumption is that “one size (cutoff) fits all”. Thus, it is not unreasonable that there will be a recognizable heterogeneity within the bins based on RACE/Demographics that would only be measurable/discernible by the granularity of deep learning. This is then complicated “downstream” by the confounding linkages of tumor stage/grade to outcome-based on access to care (ie demographics). Thus, there will be differences in histology and stage (grade+size) that will vary by time of presentation (note that tumor size will influence sampling and the degree of tumor heterogeneity recovered at the pre-analytic stage). Here in the remaining paragraph, is where I feel is the greatest impact based on demography and are major contributors to both disease outcome and what also is interpreted as tumor biology (particularly concerning size). Contributions will include known ancestry-linked differences in the frequency of TNBC (2-fold higher in blacks) which occurs at a younger age and has higher grade. Thus, all of these features of demographics can’t be disentangled from their impact on histology, particularly given the established ability of deep-learning to discern hormone receptor status. Thus, I am sure that the deep-learning signature is picking up features of age association with grade; race association with receptor status; higher grade associated with TNBC; lower age associated with TNBC; and possible differences in tumor sampling (heterogeneity) that could also be linked to differences in tumor size at presentation. All of these features are invariably linked to demographics. The quantitative (discretized) nature of this linkage to intrinsic tumor biology cannot be recognized by the pathologists without the assistance of AI, which provides the granularity that enables the quantitative distinction.

Thus the authors should take a more nuanced approach to this statement and those later derived from it in the writing. While the signature can make accurate ancestry predictions, it is not clear these predictions are un-coupled from intrinsic biology.

Thank you for bringing up this important clarification. We agree that deep learning models have the potential to identify ancestry from histology by recognizing a number of histologic features associated with ancestry, including hormone receptor status, grade, and perhaps other intrinsic factors that have not yet been described. This was actually the hypothesis that brought about the work in question, and is why ancestry takes such a central role in this paper. We initially found that accuracy for ancestry prediction often exceeded that for some plainly obvious histologic features such as grade. As we explored why ancestry prediction was so accurate, we found that prediction was heavily based on the tissue-submitting site in TCGA. For TCGA-BRCA, for example, we find that a model is trained on a mixture of all patients from all sites in TCGA, ancestry can be predicted. However, if a model is trained on TCGA patients from University of Chicago and used to predict ancestry on TCGA patients from Duke, for example, the accuracy declines substantially.

In our work, we found with standard cross validation, AUROC for ancestry prediction in TCGA BRCA was 0.798, suggesting ancestry could be predicted almost as accurately as ER status. However, with preserved site cross validation, ancestry prediction had an AUROC of 0.507, suggesting that model performance was no better than random chance. We believe preserved site cross validation more accurately portrays how models would perform in an external dataset. We hypothesize that the reason for this decline is that models were learning staining or other site specific features rather than the true histologic features that differentiate patient ancestry.

We want to avoid confusion and causing readers to draw the false conclusion that breast cancer ancestry cannot be predicted from histology – but rather reinforce that models trained in the way we have described on a very specific multisite repository will not learn the key features that distinguish cancers from different ancestries. Thus, we have reworded this section and softened our language as follows, and also provided a potential solution to allow further study of ancestry and demographic features that would avoid pollution from site-specific demographic signatures:

However, we have demonstrated that deep learning models trained on multi-site repositories such as TCGA to predict genomic ancestry have the potential to base predictions at least in part on the demographic makeup of submitting sites, rather than intrinsic tumor biology. This is evidenced by the fact that ancestry is predictable in TCGA-BRCA with standard but not preserved site cross validation. This poses a challenging ethical dilemma for the implementation of deep learning histology models. It has been well documented that women of African ancestry with breast cancer have a poorer prognosis that is not completely accounted for by stage and receptor subtype^{46,47}. Contributing factors may include delays in treatment initiation and inadequate intensity of therapy⁴⁸, and more research is needed to disentangle the biologic and non-biologic factors contributing to disparities in prognosis. As deep learning models are able to infer patient ancestry from site-specific signatures, models must be carefully implemented in an equitable fashion to avoid recapitulating the pre-existing inequities in cancer care⁴⁹. Further study within single site repositories, or repositories where tissue is stained and digitized at a single center, may promote more accurate modeling of demographic factors with deep learning.

One other concern particularly given the high percentage of patient of African ancestry submitted from one site (Chicago) is that the authors in their figures show the total number of patients contributed by each site so it is clearer what the full numeric impact of the demographic factors from this site has to the signature.

Thank you – we have updated Figure 2 and Supplemental Figure 1 to describe the number of slides contributed from submitting sites.

Finally, it might be very interesting to have the authors present confusion matrices comparing ancestry predictions from whole slide images vs genomic ancestry calls for the overall TCGA population and a few example sites. This would give readers a much more concrete idea of accuracy with which models can predict genomic ancestry

Thank you for the suggestion – we have provided confusion matrixes comparing model predictions and genomic ancestry calls for TCGA-BRCA overall and for a number of example sites in Supplemental Figure 6.